# Decoupling salinity and carbonate chemistry: Low calcium ion concentration rather than salinity limits calcification in Baltic Sea mussels

Trystan Sanders[1,2*], Jörn Thomsen[1], Jens Daniel Müller[3,4], Gregor Rehder[4], Frank Melzner[1]

[1]Marine Ecology, Helmholtz Centre for Ocean Research (GEOMAR), Kiel, Germany
[2]School of Ocean and Earth Science, National Oceanography Centre Southampton, University of Southampton, Southampton, UK*
[3]Environmental Physics, Institute of Biogeochemistry and Pollutant Dynamics, ETH Zurich, Zurich, Switzerland.
[4]Department of Marine Chemistry, Leibniz Institute for Baltic Sea Research, Warnemünde, Germany

*current affiliation

*Correspondence to*: Trystan Sanders, (t.b.sanders@soton.ac.uk)

**Abstract**

The Baltic Sea has a salinity gradient decreasing from fully marine (> 25) in the West to below 7 in the Central Baltic Proper. Habitat-forming and ecologically dominant mytilid mussels exhibit decreasing growth when salinity < 11, however the mechanisms underlying reduced calcification rates in dilute seawater are not fully understood. Both $[HCO_3^-]$ and $[Ca^{2+}]$ also decrease with salinity, challenging calcifying organisms through $CaCO_3$ undersaturation ($\Omega \leq 1$) and unfavourable ratios of calcification substrates ($[Ca^{2+}]$ and $[HCO_3^-]$) to the inhibitor ($H^+$), expressed as the extended substrate inhibitor ratio (ESIR). This study combined *in situ* monitoring of three Southwest Baltic mussel reefs with two laboratory experiments to assess how various environmental conditions and isolated abiotic factors (salinity, $[Ca^{2+}]$, $[HCO_3^-]$ and pH) impact calcification in mytilid mussels along the Baltic salinity gradient. Laboratory experiments rearing juvenile Baltic *Mytilus* at a range of salinities (6, 11 and 16), $HCO_3^-$ concentrations (300 - 2100 µmol kg$^{-1}$) and $Ca^{2+}$ concentrations (0.5-4 mmol kg$^{-1}$) reveal that as individual factors, low $[HCO_3^-]$, pH and salinity cannot explain low calcification rates in the Baltic Sea. Calcification rates are impeded when $\Omega_{aragonite} \leq 1$ or ESIR $\leq 0.7$ primarily due to $[Ca^{2+}]$ limitation which becomes relevant at a salinity of ca. 11 in the Baltic Sea. Field monitoring of carbonate chemistry and calcification rates suggest increased food availability may be able to mask the negative impacts of periodic sub-optimal carbonate chemistry, but not when seawater conditions are permanently adverse, as observed in two Baltic reefs at salinities < 11. Regional climate models predict a rapid desalination of the Southwest and Central Baltic over the next century and potentially a reduction in $[Ca^{2+}]$ which may shift the distribution of marine calcifiers westward. It is therefore vital to understand the mechanisms by which the ionic composition of seawater impacts bivalve calcification for better predicting the future of benthic Baltic ecosystems.

## 1. Introduction

### 1.1 Baltic Sea hydrochemistry

The Baltic Sea is a semi-enclosed brackish water body with a decreasing salinity (S) gradient from > 25 in the Kattegat to salinities < 3 in the Gulf of Bothnia and the Gulf of Finland (Meier, 2006; Neumann, 2010; Fig. 1). Salinity and seawater carbonate chemistry are highly variable on both spatial and temporal scales with extreme fluctuations resulting from occasional inflow events of highly saline North Sea water, wind driven upwelling of hypoxic and hypercapnic deep water, lateral transport of water masses and pronounced regional salinity gradients (Thomas and Schneider, 1999; Melzner et al., 2013; Saderne et al., 2013; Mohrholz et al., 2015). Along the Baltic Sea salinity gradient, seawater total alkalinity ($A_T$) decreases linearly from ~ 2300 µmol kg$^{-1}$ in the North Sea as the marine endmember to ~ 1600 µmol kg$^{-1}$ at a salinity of 6 in the Baltic Proper (Müller et al., 2016). In seawater that is near equilibrium with current atmospheric pCO$_2$, the dissolved bases $HCO_3^-$ and $CO_3^{2-}$ comprise ca. 96% of $A_T$. These two carbon species constitute ca. 99 % of all total dissolved inorganic carbon ($C_T$) and subsequently both $A_T$ and $C_T$ are tightly coupled (Zeebe and Wolf-Gladrow, 2007). Along with $C_T$, seawater $[Ca^{2+}]$ also decreases linearly with salinity from the North Sea to the Baltic Proper (Kremling and Wilhelm, 1997). This decreasing availability of seawater $Ca^{2+}$ and $C_T$ puts pressure on calcifying organisms that extract these 2 substrates ($Ca^{2+}$ and $C_T$) from seawater for calcification.

### 1.2 Ecophysiology of calcifying mussels

Despite reduced $[Ca^{2+}]$ and $C_T$ along the Baltic Sea salinity gradient, calcifying mussels in the genus *Mytilus* dominate the Baltic benthos particularly below salinities of 11, being the primary biogenic reef-forming species

in the central and eastern Baltic basins (Westerbom et al., 2002). In the Baltic Sea, these biogenic mussel reefs contribute more to ecosystem services than both seagrass meadows and macroalgae beds, being responsible for biodiversity hotspots and significant carbon and nutrient recycling in the benthos (Norling and Kautsky, 2008; Koivisto and Westerbom, 2010; Heckwolf et al., 2020; Attard et al., 2020). Models suggest that the contribution of filter feeding mussels to the region's biogeochemistry is so pronounced that increased bivalve aquaculture may significantly reduce levels of eutrophication in the central Baltic Sea (Kotta et al., 2020). Despite the ecological importance of Baltic *Mytilus*, calcification and growth rates of these mussels are drastically reduced below salinities of 16-11, resulting in extremely slow growing mussels and reduced maximum body size (Vuorinen et al., 2002; Riisgård et al., 2014; Sanders et al., 2018; Westerbom et al., 2019). However high abundances of mussels at salinities of 11-6 facilitate levels of ecosystem function comparable to that at higher salinities of >20 (Elmgren and Hill, 1997). Below a threshold salinity of < 5, Baltic *Mytilus* can no longer survive and ecosystem function is severely reduced (Elmgren and Hill, 1997; Vuorinen et al., 2015).

Steep declines in Baltic *Mytilus* calcification below salinities of 11 have been attributed to unfavourable protein metabolism from osmotic stress and high energetic costs of $CaCO_3$ biomineralisation related to sub-optimal seawater chemistry (Tedengren and Kautsky, 1986; Maar et al., 2015; Sanders et al., 2018; Thomsen et al., 2018). Despite the challenging environment, high levels of eutrophication resulting in increased phytoplankton food availability has been shown to mask the negative impacts of sub-optimal carbonate chemistry on calcification in Baltic *Mytilus* in Kiel Fjord (Melzner et al., 2011; Thomsen et al., 2013). Similarly, macrophyte habitats can mitigate the negative effects of seawater acidification on calcifiers by localised pH increases in adjacent seawater (Wahl et al., 2017). The complexity of environmental control on calcification in Baltic mussels makes it difficult to predict how environmental change will impact calcifying mussels and associated ecosystem services.

### 1.3 Carbonate chemistry and calcification

Since salinity, $[Ca^+]$ and $C_T$ are tightly coupled in the Baltic Sea it is difficult to decipher which of these factors primarily controls calcification. The stability of $CaCO_3$ structures in seawater is dependent on the saturation state ($\Omega$) of the $CaCO_3$ polymorph in question (predominantly aragonite or calcite in bivalves), which is defined as:

$$\Omega = \frac{[Ca^{2+}][CO_3^{2-}]}{K_{sp}} \tag{1}$$

with $K_{sp}$ being the temperature, pressure and salinity-dependent solubility product of the $CaCO_3$ polymorph under consideration. $CaCO_3$ dissolution becomes more thermodynamically favourable when $\Omega < 1$ (undersaturation), having major implications for marine organisms which biomineralize external $CaCO_3$ structures (Waldbusser et al., 2014; Ries et al., 2016). Extended periods of undersaturation of the $CaCO_3$ polymorph aragonite ($\Omega_{aragonite} \leq 1$), are commonplace in the Baltic Sea, theoretically imposing constraints on calcifying organisms that precipitate aragonite $CaCO_3$ crystals, as mytilid mussels do for production of inner shell nacre (Tyrrell et al., 2008; Melzner et al., 2011). While $\Omega_{aragonite}$ is a good predictor of biological calcification especially at fully marine conditions, there is little evidence supporting its mechanistic role in calcification physiology. Even with a rudimentary understanding of calcification mechanisms in molluscs, we know that increased $[H^+]$ inhibits extracellular calcification by diminishing the proton gradient between the slightly alkalized calcifying fluid in larvae and ambient seawater (Ramesh et al., 2017). Additionally, $HCO_3^-$ rather than $CO_3^{2-}$ is the primary source of inorganic carbon for calcification in marine organisms (Bach, 2015). This is because in seawater at equilibrium with current atmospheric pCO$_2$, $HCO_3^-$ is ca. 10-fold more abundant in seawater than $CO_3^{2-}$ and $HCO_3^-$ transporters have been localised in the calcifying tissues of marine organisms (Roleda et al., 2012; Zoccola et al., 2015; Fassbender et al., 2016; Hu et al., 2018). Since the availability of $HCO_3^-$ stimulates calcification, a more accurate and mechanistically relevant predictor for calcification is the substrate ($HCO_3^-$) inhibitor ($H^+$) ratio or SIR (Jokiel, 2013; Bach, 2015; Thomsen et al., 2015; Cyronak et al., 2016). The effects of $[Ca^{2+}]$ are generally excluded in calculating SIR (Eq. 2) as $[Ca^{2+}]$ is high and generally constant at oceanic salinity (~ 10 mmol kg$^{-1}$).

$$SIR = \frac{[HCO_3^-]}{[H^+]} \tag{2}$$

At salinities < 8 however, limitation of seawater $Ca^{2+}$ has been shown to reduce calcification in larval Baltic *Mytilus* (Thomsen et al., 2018). Therefore, adopting an extended SIR (ESIR) to include $[Ca^{2+}]$, may provide an even more accurate predictor of calcification under low salinity conditions.

$$ESIR = \frac{[Ca^{2+}][HCO_3^-]}{[H^+]} \tag{3}$$

Previous work reveals that calcification in Baltic larval bivalves drops significantly when ESIR ≤ 0.7 (Thomsen et al., 2018). Application of the ESIR is particularly important in coastal and estuarine zones such as the Baltic, where freshwater input, photosynthetic activity and wind-driven upwelling can delineate carbonate system parameters (pH and $pCO_2$) from salinity and weaken the predictive power of SIR for calcification (Melzner et al., 2013; Fassbender et al., 2016). Severe desalination and warming are predicted in the Baltic Sea over the next 50-100 years, due to changes in weather patterns and precipitation, yet our ability to predict future carbonate chemistry changes remains somewhat limited due to potential antagonistic effects of desalination and continental weathering (Gräwe et al., 2013; Müller et al., 2016). The combination of our poor understanding of 1) how Baltic Sea carbonate chemistry will change in the future and 2) how environmental factors individually control calcification, make it difficult to predict the impacts of environmental change on Baltic Sea calcifiers and by extension, Baltic Sea ecosystems.

**1.4 Scope of this study**

This study addresses the impacts of environmental change on Baltic Sea calcifying mussels, by combining a multi-year environmental monitoring programme and field growth study and at three Baltic *Mytilus* reefs and two complementary laboratory experiments. Field salinity, food availability, carbonate chemistry and calcification rates were monitored in three mussel reefs along the Southwest Baltic Sea salinity gradient. This was complemented by laboratory experiments in which salinity was decoupled from calcification substrate availability ($HCO_3^-$ and $Ca^{2+}$) and pH to investigate how these individual factors control calcification rate. The results from this study shed light on the primary environmental factors driving growth rates in ecologically dominant calcifying Baltic mussels. This will support biogeochemical model projections to better predict changes in distribution and function of Baltic calcifying bivalves in the future.

**2. Material and Methods**

**2.1 Environmental monitoring**

Three monitoring sites were selected along the Southwest Baltic Sea coast based on previous population comparison studies spanning the geographic range of the steepest section of the Baltic Sea salinity gradient (Meier, 2006; Sanders et al., 2018; Fig. 1). These sites were: Kiel (54° 19' 49" N, 10° 8' 60" E), Ahrenshoop (54° 23' 7" N, 12° 25' 24" E) and Usedom (54° 3' 21" N, 14° 0' 40" E). Mini CTD (conductivity, temperature, depth) loggers (Star-Oddi, Iceland) logging data every 3 h, were deployed on wooden pillars at each site between July and Sept. 2015 (0.5-1.5 m depth) and replaced every 2-3 months until Jan. 2018. Field $[Ca^{2+}]$ was calculated from salinity values using $[Ca^{2+}]$-chlorinity relationships from Kremling and Wilhelm, 1997 and a chlorinity-salinity relationship from Millero, 1984. Reference measurements from field water samples concurred with these calculations from the literature (Fig. S1). Additionally, every 2-3 months, two 500 ml water samples were taken from Ahrenshoop and Usedom (1 m depth) and immediately poisoned with saturated $HgCl_2$ solution (0.5 ‰ final concentration) for $pH_{total}$, $A_T$ and $C_T$ measurements at the Leibniz Institute for Baltic Sea Research, Warnemünde. Kiel carbonate chemistry data was taken from published monitoring data for Jan.-Dec. 2015 (Hiebenthal et al., 2017). Water $pH_{total}$ was measured spectrophotometrically at 25 °C with non-purified m-cresol purple as indicator dye (pH calculated according to Müller and Rehder, 2018, dye pH-perturbation corrected after Hammer et al., 2014). $C_T$ was determined with a SOMMA system (Single Operator Multi-Parameter Metabolic Analyzer) operated at 15°C and $A_T$ was determined by open-cell titration at 20 °C (Dickson et al., 2007). Field carbonate chemistry parameters were calculated using $C_T$ and $pH_{total}$ as inputs using the CO2Sys_v2.1 program ($KHSO_4$ constants from Dickson, 1990, K1 and K2 dissociation constants from Millero, 2010).

**2.2 Field calcification**

In mussel populations originating from the three monitoring sites chosen in this study, calcification rates have been shown to drop significantly below salinities of 11 (Sanders et al., 2018). To investigate the underlying cause of this in the field, mussel settlement structures (n = three per site) were placed at each site in March 2016 to monitor the growth of the spring cohort of settled juveniles over the course of 15 months. The structures were made from a 35 cm long cylinder of grey PVC pipe with a diameter of 10 cm. Twenty x 2 cm diameter holes were drilled into the sides and a 0.2 mm gridded nylon net (standard seed sock for mussel aquaculture) was placed diagonally across the inside to increase surface area allowing for maximum settlement (Fig. S2). These settlement structures were cable-tied to wooden pillars at each site at ca. 0.5-1 m depth and samples of 50 juvenile mussels were taken for shell length measurements from each site every 2-3 months. Individual shell mass (SM, µg $CaCO_3$) was calculated from individual shell lengths using specific SL-$CaCO_3$ relationships developed for each of the three populations (Sanders et al., 2018; Fig. S3). To allow direct comparison between sites, individual calcification rates (µg $CaCO_3$ $d^{-1}$) are expressed as the linear slopes of $CaCO_3$ mass over time (days) over the first three sampling time points before growth rates deviate from linear.

**2.3 Food availability**

Chlorophyll-*a* (chl-*a*) concentrations (µg L$^{-1}$) were used as a proxy for phytoplankton food availability in Baltic Sea surface waters (Wasmund et al., 2011). Values for chl-*a* concentrations at the three monitoring sites were obtained from field monitoring data supplied by the Landesamt für Landwirtschaft, Umwelt und ländliche Räume (LLUR) sampled from Mönkeberg (54° 21' 7" N, 10° 10' 36" E), 2.5 km from the field monitoring site in Kiel Fjord. Chl-*a* monitoring data for Ahrenshoop and Usedom were supplied by the Landesamt für Umwelt, Naturschutz und Geologie (LUNG) sampled from near Fischland (54° 20' 43" N, 12° 26' 55" E), 1 km from the sampling site in Ahrenshoop and Zinnowitz (54° 4' 57" N, 13° 54' 57" E), 4 km from the monitoring site in Usedom.

**2.4 Laboratory experiments: animal collection and maintenance**

A genetic hybrid zone and gradient in allele frequencies exists in the Southwest Baltic Sea from Baltic *Mytilus edulis* dominated hybrids at higher salinities (> 16) to Baltic *Mytilus trossulus* hybrids at low salinities (< 8), with a genetic transition zone occurring at salinities of 10-12 (Stuckas et al., 2017). Two cohorts of freshly settled Baltic *Mytilus* from the genetic transition zone (mean shell length, SL = 0.6 mm - cohort 1, and 3.0 mm - cohort 2) were collected from wooden pillars in Ahrenshoop, Germany (54° 23' 13" N, 12° 25' 37" E., mean salinity = 10.8, temperature = 19.3 °C, depth < 1 m) in July 2014 and 2016 (Fig. 1). Animals were transported within 6 h in chilled, aerated cool boxes to GEOMAR, Kiel, stored in 20 L plastic acclimation aquaria (16 °C) for three weeks prior to experiments and fed twice daily with live *Rhodomonas salina* to achieve phytoplankton concentrations of > 10 000 cells ml$^{-1}$ which is considered saturated food availability for Baltic *Mytilus* (Riisgård et al., 2013). Water was exchanged in acclimation aquaria every three days and the correct salinity was obtained by mixing 0.22 µm filtered seawater (FSW) from Kiel Fjord (salinity ~ 16) with distilled water and adjusting $A_T$ to site-specific values via addition of 1M NaHCO$_3$ solution.

**2.5 [HCO$_3^-$] and [Ca$^{2+}$] manipulation experiments**

Two laboratory experiments were conducted to study the impacts of calcification substrate availability (HCO$_3^-$ and Ca$^{2+}$) on calcification. The first experiment exposed Baltic *Mytilus* juveniles (cohort 1, mean SL = 0.6 mm) to three salinities (16, 11 and 6, reflecting mean salinities at the three monitoring sites) and six bicarbonate ion concentrations (300, 600, 900, 1500, and 2100 µmol kg$^{-1}$), reflecting $A_T$ of full-strength seawater down to salinities < 3 in the northern Baltic Sea (Müller at al., 2016). The experiment was conducted in 2 L plastic aquaria each containing ~ 1600 juvenile mussels with four replicate aquaria per treatment (15 treatments, 60 aquaria in total). Experimental aquaria were continuously aerated with pressurised ambient air throughout the 70-day experimental duration. Salinity and carbonate chemistry manipulation was carried out by diluting FSW from Kiel Fjord to experimental salinities (16, 11 and 6) with distilled water in 800 L containers with $C_T$ and pH$_{NBS}$ determined weekly in this stock FSW (Sect. 2.6). To achieve experimental [HCO$_3^-$] values, 1M NaHCO$_3$ or 1M HCl was added to increase or decrease [HCO$_3^-$], respectively (mean water parameter values in experimental aquaria for the duration of the experiment are given in Table 2). Water was prepared individually for each treatment aquarium 24 h before water changes and equilibrated with pressurized air at atmospheric pCO$_2$. Water changes were conducted twice weekly for the first 30 days and three times weekly for the latter 40 days to prevent significant deviations in target water chemistry conditions as animals grew. Dead animals were removed and counted during water changes with declines in animal abundances accounted for when calculating volume of food addition. $C_T$ and pH$_{NBS}$ were determined in experimental aquaria immediately before and after water changes to ensure that [HCO$_3^-$] and pH did not deviate by > 200 µmol kg$^{-1}$ or > 0.1 units, respectively, due to biological activity or the addition of phytoplankton food culture. Phytoplankton food was added twice daily during the first 30 days and three times daily during the latter 40 days to compensate for increasing biomass ensuring saturated availability of food during experiments.

A second experiment was conducted to investigate the impact of [Ca$^{2+}$] on calcification rates. This experiment utilised the same experimental design as the first experiment, exposing juvenile Baltic *Mytilus* (cohort 2, mean SL = 3.0 mm) to the same three salinities (16, 11 and 6) and five different [Ca$^{2+}$] values (0.5, 1, 2, 3, 4 mmol kg$^{-1}$) covering the natural range of [Ca$^{2+}$] in the Southwest and Central Baltic (Thomsen et al., 2018). Experimental seawater was manipulated through the preparation of Ca$^{2+}$ free artificial seawater (CFASW) in three stock solutions (salinity 6, 11 and 16) by adding NaCl, NaSO$_4$, KCl, NaHCO$_3$, KBr, H$_3$BO$_3$, MgCl$_2$ and SrCl$_2$ to deionised water (Table S1; Kester et al., 1967). Aquarium volumes in the Ca$^{2+}$ experiment could not be as high as in the HCO$_3^-$ experiment due to an inability to produce large volumes of CFASW. Consequently, the Ca$^{2+}$ experiment was conducted in smaller 50 ml plastic aquaria with two animals per tank and four replicates (15 treatments, 60 aquaria in total) and lasted 37 days. Fewer animals were used per aquarium to minimise the impact of biomineralisation on seawater carbonate chemistry due to smaller water volumes and larger experimental animals (Table S2). No mortality was observed in any aquaria during this experiment. Stock CFASW was prepared 24 h before water changes and continuously aerated with pressurised ambient air to equilibrate water with atmospheric pO$_2$ and pCO$_2$. For preparation of experimental water, [Ca$^{2+}$] was adjusted through the addition

of a 1M CaCl₂ stock solution to achieve target $Ca^{2+}$ concentrations. $C_T$ and $[Ca^{2+}]$ were measured twice weekly in stock CFASW (Sect. 2.6). $[Ca^{2+}]$ was measured using a flame photometer (EFOX 5053, Eppendorf) calibrated with urine standards (Biorapid; see Table 3 for mean water parameters in experimental treatments for the duration of experiment). Aquaria were not aerated during experimentation but twice weekly monitoring of $pH_{NBS}$ and $C_T$ before and after water changes, revealed mean pH did not change by more than 0.1 pH units and $[Ca^{2+}]$ did not deviate by more than 0.2 mmol kg⁻¹. Twice weekly water changes were sufficient to prevent respiratory build-up of $CO_2$ between water changes (Table 2 & 3) and animals were fed once daily with *R. salina* to ensure comparable levels of food per unit body mass between both experiments.

While it may be argued that the disparity in aquarium volumes and animal numbers makes comparisons between both experiments more difficult, measures were taken to ensure comparable conditions between both experiments (Table S2). Both laboratory experiments used juvenile mussels from the same sample population at the same developmental stage. Regular monitoring of carbonate chemistry in experimental aquaria ensured the frequency of water changes was sufficient to minimise the impacts of both biological activity and phytoplankton food culture addition on target carbonate chemistry values. Live microalgal food (*R. salina*) was added at pre-determined frequencies in both experiments to maintain saturated feeding conditions and a comparable level of daily food ration (no. of phytoplankton cells mg⁻¹ body mass) between both experiments (Table S2). As biomass changed throughout the course of both experiments either through growth or mortality, feeding frequency was adjusted to ensure comparable concentrations of phytoplankton cells mg⁻¹ body mass (Table S2) across both experiments and between treatments. Cell concentrations in phytoplankton cultures were monitored daily using a Multisizer 3 Coulter Counter (Beckman, Germany). In both experiments, aquaria were held in 16 °C water baths and mean body mass L⁻¹ in experimental aquaria (average over duration of experiments) was comparable between both experiments (Table S2) minimising the differential impacts of biological activity on experimental seawater conditions. Finally, unlike many other studies, after dilution of FSW with distilled water, experimental $A_T$ was adjusted to site-specific levels across all salinities to accurately imitate natural seawater conditions present in low salinity (< 11) Baltic Sea habitats.

### 2.6 Carbonate chemistry
For carbonate chemistry measurements in experiments at GEOMAR, $pH_{NBS}$ was measured three times weekly using a WTW pH 3110 probe at 25 °C calibrated with two standard precision NBS buffers (Radiometer analytical) at pH 7.0 and 10.0. Experimental temperature and salinity were monitored daily using a WTW Cond 315i probe. $C_T$ was analysed using an AIRICA $CO_2$ analyser (Marianda, Kiel, Germany) calibrated by measuring certified reference material (Dickson et al., 2003). Experimental seawater carbonate system parameters ($A_T$, $[H^+]$, $[HCO_3^-]$, $[CO_3^{2-}]$, and $\Omega_{aragonite}$) were calculated with $C_T$ and $pH_{NBS}$ as inputs using the same methods described in Sect. 2.1. Values for $\Omega_{aragonite}$ in the calcium experiment were calculated according to Eq. (1) from measured $[Ca^{2+}]$ in each technical replicate, as well as calculated $[CO_3^{2-}]$ and $K_{sp}$. The SIR and ESIR were calculated according to Eq. (2) and Eq. (3).

### 2.7 Experimental calcification rates
Individual calcification rates in laboratory experiments were determined by calculating the change in $CaCO_3$ mass (a proxy for shell mass; SM) over time between two time points ($t_0$ and $t_1$). Individual SM (mg) was calculated separately for each experiment at the beginning of experiments ($t_0$) from mean SL per replicate tank and using a population specific SL-$CaCO_3$ mass relationship for the experimental population (Sanders et al., 2018; Fig. S3). Individual SM at the termination of each experiment ($t_1$) was calculated by removing body tissue from individuals using forceps under a stereomicroscope and placing empty shells individually into pre-weighed and pre-dried aluminium foil cups which were then placed in a muffle furnace and ashed at 450 °C for 4 h to remove all organic content. The remaining inorganic mass was used as a proxy for $CaCO_3$ mass. Individual calcification rates were calculated with the following equation:

$$\text{Calcification rate} \left( \mu g\ CaCO_3\ d^{-1} \right) = \frac{(SM)t_1 - (SM)t_0}{\text{no.of days}} / 1000 \tag{4}$$

Where SM = shell mass (mg) at time points $t_0$ and $t_1$ represent SM at the beginning and end of each experiment, respectively and no. of days represents the duration of the experiments; 70 and 37 days for the $HCO_3^-$ and $Ca^{2+}$ experiments, respectively.

### 2.8 Data analysis
All data analysis was conducted using R Software version 3.6.3 (R Core Team, 2020) with all packages used listed in Table S3. Data distributions were tested for normality (Shapiro-Wilk test) and homogeneity of variances (Levene's test) before running parametric tests. In the bicarbonate experiment, a negative exponential decay model

was fit to the relationship between [$HCO_3^-$] and calcification rate using a nonlinear least squares method for parameter optimisation. The Akaike Information Criterion (AIC) was used to designate the most parsimonious model (Table S5). Linear regression models were fit to calcification rates with [$Ca^{2+}$] as a co-variate and salinity as a factor in the calcium experiment. Negative exponential decay models were fit to explain the relationship between ESIR and $\Omega_{aragonite}$ on laboratory calcification rates with model parameters ($C_{max}$ and $K$) compared (showing 95 % confidence intervals) between ESIR and $\Omega_{aragonite}$ as predictors of calcification. The residual sum of squares (RSS) was used as a measure of model fit. The impacts of salinity were analysed statistically by comparing the $C_{max}$ parameter (± 95 % confidence interval) of the non-linear model in the bicarbonate experiment and ANCOVA in the calcium experiment. Field calcification rates (monthly means) and environmental parameters (salinity, pH, chl-$a$, [$Ca^{2+}$], [$HCO_3^-$], $\Omega_{aragonite}$, and ESIR) were analysed using parametric tests (ANCOVA) with time as a co-variate for field calcification rates and site as a fixed factor. Pairwise comparisons were conducted on significant factors by way of a Tukey test. Field temperatures for the given monitoring period (2015-2017) were analysed non-parametrically using a Kruskall-Wallis test. Finally, field calcification rates were calculated as the values for the linear slopes of cumulative calcification over time for the first three sampling periods. Log-transformed mean calcification rates for each site were plotted against mean values for individual field environmental parameters.

## 3. Results

### 3.1 Environmental monitoring

Salinity was different between all three monitoring sites (ANCOVA, $F_{(2, 17253)} = 38518$, $P < 0.001$, Table S6) being highest and more variable in Kiel and lowest and the most stable in Usedom (Fig. 2). As salinity and [$Ca^{2+}$] exhibit a linear relationship (Fig. S1), [$Ca^{2+}$] also differed to the same degree as salinity between the three sites (Fig.2). Calculated mean [$HCO_3^-$] at the three field sites (Table 1) were significantly higher in Kiel compared to Ahrenshoop and Usedom (ANOVA, $F_{(2, 55)} = 38.80$, $P < 0.001$, Table S6) with values never dropping below 1600 µmol kg$^{-1}$ at any of the three sites during the monitoring period (Fig. 3c). While pH$_{total}$ was highly variable, particularly in Kiel (Fig. 3a), no difference was observed between all 3 sites (ANOVA, $F_{(2,55)} = 1.217$, $P = 0.304$). Salinity and $A_T$ exhibited a linear relationship similar in slope to the S-$A_T$ relationship reported by Müller et al., 2016 for the Kattegat in 2014 (Fig. 4). However, $A_T$ values measured in this study were ca. 100 µmol kg$^{-1}$ higher at a given salinity than those reported by Müller et al., 2016 for the Kattegat and Central Baltic Proper (model parameters given in Table S7). Calculated $\Omega_{aragonite}$ was higher in Kiel compared to the other two sites (ANOVA, $F_{(2, 55)} = 7.22$, $P = 0.002$, Fig. 3e) with periods above and below $\Omega_{aragonite} = 1$ in Kiel (mean = 1.17) and mean values below saturation ($\Omega_{aragonite} < 1$) in Ahrenshoop or Usedom at 0.83 and 0.76, respectively (Fig. 3e, Table 1). Similar patterns where observed in the calculated values for ESIR at the three sites (Fig. 3g) with Kiel exhibiting significantly higher mean ESIR values than both Ahrenshoop and Usedom (ANOVA $F_{(2, 55)} = 10.88$, $P < 0.001$), with a mean value of 1.05 ± 0.6 (Table 1). Mean ESIR values were below the threshold of 0.7 proposed by Thomsen et al., 2018 in Ahrenshoop and Usedom (means of 0.65 ± 0.19 and 0.55 ± 0.26, respectively), although periods above and below this threshold were observed at all sites (Fig. 3g). Mean chl-$a$ concentrations at each site (Fig. 3i, Table 1) derived from monitoring data were significantly lower at Ahrenshoop compared to Kiel and Usedom (ANOVA, $F_{(2, 77)} = 13.8$, $P < 0.001$, Table S6) and not significantly different between Kiel and Usedom (Tukey $post-hoc$, $P = 0.35$).

### 3.2 Calcification rates

Calcification rates in the bicarbonate experiment exhibited a significant negative exponential decay relationship with decreasing [$HCO_3^-$] (Table S8), with calcification rates decreasing most abruptly below ca. 1000 µmol kg$^{-1}$ (Fig. 5a) at salinities of 11 and 16. Maximum calcification rates ($C_{max}$) at salinity 6 were only ~ 13 % of those at 11 and 16 and not different between salinities of 11 and 16 (Fig. 5c). The subsequent experiment which decoupled [$Ca^{2+}$] from salinity, revealed a significant linear decrease (parameters given in Table S8) in calcification rate with [$Ca^{2+}$] (ANCOVA, $F_{(2,54)} = 106.9$, $P < 0.001$) across the range of experimental [$Ca^{2+}$] (Fig. 5b). Salinity as a single factor had no significant impact on calcification rates (ANCOVA, $F_{(2,54)} = 0.83$, $P = 0.442$) however, a significant interaction effect between salinity and [$Ca^{2+}$] suggests that the effects of [$Ca^{2+}$] on calcification rate vary at different salinities. Combining both experiments, calcification rates did not exhibit any significant correlation with [$Ca^{2+}$] or [$HCO_3^-$] alone under the given conditions (Fig. S4, Table S9). Calcification rates plotted separately against $\Omega_{aragonite}$ and ESIR across both experiments revealed statistically significant estimations for both parameters ($C_{max}$ and $K$) in the negative exponential decay model (Fig. 6, Table S6). Estimated model parameters were not significantly different between $\Omega_{aragonite}$ nor ESIR as calcification predictors, (95 % CI; Fig. S6). Although, a lower residual sum of squares (residual SS = 2866) in the ESIR model compared to the $\Omega_{aragonite}$ model (residual SS = 3159) indicates slightly better performance of the ESIR model as a predictor for calcification in the Southwest and Central Baltic Sea. In the bicarbonate manipulation experiment, mean experimental [$HCO_3^-$] values were within 200 µmol kg$^{-1}$ of target values except for 1 out of 15 treatments (Table 2). In the calcium manipulation experiment, [$Ca^{2+}$] was within 0.2 mmol kg$^{-1}$ of target values with the exception of 2 out of 15 treatments (Table

3). Despite different aquarium volumes and animal number per aquarium, mean biomass $L^{-1}$ in experimental aquaria in both experiments were comparable at 24.1 and 13.2 mg $L^{-1}$ for the bicarbonate and calcium ion manipulation experiments, respectively (Table S2). Mean number of phytoplankton cells per unit biomass were also comparable between both experiments (Table S2). Despite experimental aquaria not being aerated during the calcium manipulation experiment, mean pH values fluctuated by 0.04 - 0.12 units between water changes and were comparable to field pH values at all 3 monitoring sites (Tables 1 and 3) suggesting minimal impacts of animal respiration on carbonate chemistry during the experiment. Calcification rates were comparable between both laboratory experiments (Fig. 5) and with field calcification rates, at least for Ahrenshoop (mean salinity: 10.9) and Usedom (mean salinity: 7.0) populations (Table 1). Mussels from Kiel (mean salinity: 15.2) exhibited significantly higher calcification rates (ANCOVA, $F_{(2,12)} = 570$, $P < 0.001$) compared to the two other field sites, being 1-2 orders of magnitude higher (Fig. 7). This contrasts with measured laboratory calcification rates where salinities 16 and 11 treatments exhibited comparable calcification rates ($\mu$g $CaCO_3$ $d^{-1}$).

## 4. Discussion
### 4.1 Calcification in the field
The aim of this study was to identify the primary abiotic drivers responsible for decreasing calcification rates in mussel reefs along the Baltic Sea salinity gradient. Our results revealed both ESIR and $\Omega_{aragonite}$ values were almost permanently below critical thresholds and saturation levels, respectively, in Ahrenshoop and Usedom and extended periods of undersaturation were observed in Kiel Fjord. Yet despite these conditions, net positive calcification rates were recorded at all sites with mussels in Kiel Fjord exhibiting calcification rates 1-2 orders of magnitude higher than the two low salinity sites. This is interesting, as extremely low and variable pH (Fig. 3b) and other carbonate chemistry parameters have been well documented in Kiel Fjord with extended periods of $\Omega_{aragonite} < 1$ (undersaturation), particularly in early autumn during seasonal upwelling of hypercapnic water (Thomsen et al., 2010; Melzner et al., 2013; Saderne et al., 2013). These drastically higher rates of calcification in Kiel Fjord are unlikely to result solely from high ESIR and $\Omega_{aragonite}$ values, as the laboratory experiments in this study clearly reveal minor differences in calcification rates above and below ESIR saturation (Fig. 6), compared to field calcification rates. It is more probable that the high levels of eutrophication in Kiel Fjord and consequently high phytoplankton food concentrations are responsible for the rapid calcification observed in this study (Melzner et al., 2011; Thomsen et al., 2013). Chl-$a$ monitoring data in this study partially supports this with mean values in Kiel significantly higher than those at Ahrenshoop. However, chl-$a$ values in Usedom (low salinity) were comparable with those measured in Kiel despite drastically lower calcification rates in Usedom. Two-fold differences in particulate organic carbon (POC) have been documented between the inner and outer Kiel Fjord which cause higher food availability in the inner Fjord and thereby facilitate higher calcification rates (Thomsen et al., 2013). The growth monitoring site of this study was also located in the inner Fjord, however chl-$a$ measurements were taken in the outer Fjord and subsequently may not be fully representative of the Kiel mussel monitoring site. Values for chl-$a$ and food availability to inner Kiel Fjord mussels may in fact be significantly higher than reported here, which likely explains the extremely high growth rates in Kiel compared to the other two monitoring sites. Conversely, the fact that calcification rates in Usedom were considerably lower than Kiel is likely due to limited $[Ca^{2+}]$ and sub-optimal ESIR values. The laboratory experiments presented here indicate that calcification rates decrease linearly when $[Ca^{2+}] < 4$ mmol $kg^{-1}$ and monitoring data revealed mean $[Ca^{2+}]$ at Ahrenshoop and Usedom, but not in Kiel, to be well below 4 mmol $kg^{-1}$ (Fig. 2). Although pH is intrinsically linked to both $\Omega$ and ESIR, mean $pH_{total}$ values were not different between the 3 sites (Fig. 3a) despite significant differences between $\Omega$ and ESIR. $[HCO_3^-]$ was ~ 10 % lower in Ahrenshoop and Usedom compared to Kiel, whereas $[Ca^{2+}]$ was 22 % and 48 % lower in Ahrenshoop and Usedom than in Kiel, respectively. This suggests that low calcification rates at salinities $\leq$ 11 in the Southwest Baltic result from low ESIR or $\Omega$ values primarily due to low seawater $Ca^{2+}$ availability, rather than high $[H^+]$ and low $[HCO_3^-]$ (Fig. S9).

### 4.2 Carbonate chemistry
This study aimed to investigate the mechanisms by which low salinity negatively impacts calcification in Baltic mussels. Calcification relies on the uptake of both calcium and inorganic carbon from seawater to precipitate $CaCO_3$ crystals, and in the low saline Southwest Baltic Sea, the availability of both substrates is significantly lower than in the North Sea (Kremling and Wilhelm, 1997; Müller et al., 2016). The laboratory experiments presented in this study demonstrate that $HCO_3^-$ availability begins to limit calcification below ~1000 $\mu$mol $kg^{-1}$ (Fig. 5a), in line with similar experiments on *Mytilus edulis* juveniles (Thomsen et al., 2015). This corresponds to a $A_T$ of ~ 1050 $\mu$mol $kg^{-1}$ (at the experimental pH of 7.71). Such low $[HCO_3^-]$ are unlikely under fully marine conditions, due to high and stable $A_T$ in ocean waters of ~ 2300 $\mu$mol $kg^{-1}$ (Millero et al., 1998). However, estuarine and brackish water environments are often characterised by low $A_T$ due the linear relationship between salinity and $A_T$, and characteristically have a weaker pH buffering capacity and lower availability of inorganic carbon (Miller et al., 2009). Despite the low salinity, the Baltic Sea regionally exhibits relatively high $A_T$ due to high riverine $A_T$ from the southern drainage basin (Beldowski et al., 2010; Müller et al., 2016). Monitoring results from

three coastal sites in the Southwest Baltic revealed higher values of $A_T$ for any given salinity, when compared to the 2014 S-$A_T$ relationship presented in Müller et al., 2016 (Fig. 4). As samples were taken in direct proximity to the coastline, this is likely due to the influence of local freshwater endmembers (either rivers or submarine groundwater discharge). Local freshwater endmembers in the southern drainage basin are known to have a higher $A_T$ than the mean volume-weighted Baltic Sea freshwater endmember, which is reduced by the influence of low $A_T$ rivers in Scandinavia (Müller et al., 2016). The proximity to the southern coastline and higher fraction of freshwater from the southern drainage basin likely explains the offset of the $A_T$-S relationships from our monitoring sites with respect to the relationships reported by e.g Müller et al., 2016. Dissolved organic carbon (DOC) species also constitute 1.5-2.3 % of $A_T$ in the Southwest and Central Baltic, and DOC from the Oder river discharge may contribute to the high coastal $A_T$ observed in the Southwest Baltic in this study (Kuliński et al., 2014). The high $A_T$ observed across all sites in this study (1856-2071 µmol kg$^{-1}$) suggests that HCO$_3^-$ limitation at $A_T$ values < 1050 µmol kg$^{-1}$ is likely not the primary reason for reduced calcification in Baltic mussels.

It is not possible to disentangle the effects of inorganic carbon availability from pH in the bicarbonate limitation experiment. This is because the methodologies applied in this study of manipulating [HCO$_3^-$] meant reducing $A_T$ in parallel and consequently lowering seawater pH as pCO$_2$ remained unchanged (Table 2). Previous experiments utilising methods which manipulate individual parameters of the carbonate system have suggested pH as a sole factor does not correlate strongly with molluscan calcification when compared to SIR, [CO$_3^{2+}$] and $\Omega_{aragonite}$ (Waldbusser et al., 2014; Thomsen et al., 2015). Reduced seawater pH through experimentally increasing pCO$_2$, has been demonstrated to reduce calcification rates in bivalve larvae and coral due to a reduction in the proton gradient between ambient seawater and the calcifying space leading to reduced $\Omega_{aragonite}$ at the site of calcification (Allison et al., 2014; Waldbusser et al., 2014; Ramesh et al., 2017). However, it is not possible to differentiate between the impacts of ambient seawater $\Omega$, or alterations in carbonate chemistry at the site of calcification on net CaCO$_3$ precipitation/dissolution in marine bivalves, since many marine organisms actively modify the carbonate chemistry at the site of calcification (Allison et al., 2014; Cyronak et al., 2016; Ramesh et al., 2017). Calcification is stimulated by the availability of calcification substrate (Ca$^{2+}$ and HCO$_3^-$) and inhibited by H$^+$, therefore the application of the SIR provides a more physiologically oriented predictor of calcification, than $\Omega$ alone. However, the nature of the carbonate system results in the SIR co-correlating strongly with both [CO$_3^{2+}$] and $\Omega$ at a constant salinity, temperature and pressure (Bach, 2015; Thomsen et al., 2015; Cyronak et al., 2016; Fassbender et al., 2016).

In our experiments SIR ([HCO$_3^-$]/[H$^+$]) proved to be a poor predictor of calcification (Fig. S5) in comparison to $\Omega_{aragonite}$ (Fig. 5a) when [Ca$^{2+}$] was experimentally manipulated. Building on the findings from the HCO$_3^-$ limitation experiment, this second experiment individually isolated the impacts of salinity and [Ca$^{2+}$] revealing that salinity as a sole factor had no significant impact on calcification rates, whilst [Ca$^{2+}$] correlated linearly with calcification across all [Ca$^{2+}$]. This suggests Ca$^{2+}$ to be the limiting factor for calcification at concentrations < 4 mmol kg$^{-1}$ or corresponding to a salinity of ~ 11 on the natural salinity gradient, in line with previous studies on Baltic mussel larvae and adults (Kossak, 2006; Sanders et al., 2018; Thomsen et al., 2018). Negative exponential decay models did not return significant parameter estimations for $C_{max}$ (maximum calcification rates) in the calcium experiment as in the bicarbonate experiment (Table S8). This indicates that threshold saturation values of [Ca$^{2+}$] are above the maximum experimental levels tested here (ca. 3.8 mmol kg$^{-1}$). A threshold of ~ 4 mmol kg$^{-1}$ for calcification has been observed for larvae and seems to be comparable for juveniles as well (Thomsen et al., 2018). Further experiments spanning a wider range of [Ca$^{2+}$] eg. 0.5-8 mmol kg$^{-1}$, may reveal significant effects of salinity on calcification when [Ca$^{2+}$] > 4 mmol kg$^{-1}$. As a single factor, [Ca$^{2+}$] was a poor predictor of calcification across both laboratory experiments (Fig. S4b). However, in this study, extending the SIR by including [Ca$^{2+}$] as a substrate (ESIR) yielded a significant relationship with calcification, in line with $\Omega_{aragonite}$ (Fig. 5). There was no statistically significant difference between both ESIR and $\Omega_{aragonite}$ models as calcification predictors in this study which may result from the low values of both in the Baltic Sea. $\Omega_{aragonite}$ is particularly low in winter when low temperatures (Fig. S7) and high pCO$_2$ from upwelling and water mixing, increase $K_{sp}$ and decrease [CO$_3^+$], respectively (Eq. 1). Salinity and temperature also impact aragonite saturation state ($\Omega_{aragonite}$) due to the salinity and temperature dependant nature of the solubility product of aragonite, $K_{sp}$ (Eq. 1) and the CO$_2$ dissociation constants. Conversely, temperature has almost no impact on ESIR when salinity and pressure are constant (see Bach, 2015). ESIR is also unimpacted by salinity *per se*, but rather [Ca$^{2+}$] which correlates linearly with salinity in the Baltic Sea. Although future trends in the S-Ca$^{2+}$ relationship in the Southwest and Central Baltic may deviate from current relationships (Sect. 4.5). Both ESIR and $\Omega$ generally correlate well in marine systems, however this linear relationship is skewed in coastal zones where variability in carbonate chemistry parameters is high (Fassbender et al., 2016). This is highlighted in Fig. S8, where a given $\Omega_{aragonite}$ of 1.0 can correspond to ESIR values of between 0.5-1.3 (well above and below the limiting ESIR threshold of 0.7). ESIR and $\Omega_{aragonite}$ are certainly stronger predictors of calcification than calcification substrate availability alone in the

Southwest Baltic. However, the strong interdependency between both parameters makes it impossible to differentiate one from the other as a more powerful predictor of calcification in this study.

### 4.3 Salinity and calcification

Salinity is known to have a strong impact on bivalve growth and survival in the Baltic Sea, drastically reducing growth rates at salinities $\leq 11$ (Kautsky et al., 1990; Kossak, 2006: Riisgård et al., 2014; Sanders et al., 2018). It has long been assumed that the underlying physiological mechanism of this stems from intracellular osmotic stress at low salinities and inefficient protein metabolism (Tedengren and Kautsky, 1986; Maar et al., 2015). However recent work suggests that a reduced ability to biomineralize $CaCO_3$ may be limiting growth rates at low salinities, rather than inefficient growth associated with osmotic stress (Riisgård et al., 2014; Sanders et al., 2018; Thomsen et al., 2018; Sillanpää et al., 2020). This is the first study to decouple salinity from both $[HCO_3^-]$ and $[Ca^{2+}]$ and empirically test the impacts of each individual factor on bivalve calcification. In line with the findings of previous studies, both field monitoring and the bicarbonate laboratory experiment show that at salinities $< 11$, calcification rate is significantly reduced. However, when decoupled from $[Ca^{2+}]$ in the calcium experiment, salinity as a single factor did not significantly impact calcification. Interestingly, calcification rates were significantly reduced at a salinity of 6 in the bicarbonate experiment despite salinity having no impact on calcification in the calcium experiment. This could be a result of maximum $[Ca^{2+}]$ in the calcium experiment ($\sim 3.8$ mmol kg$^{-1}$) being slightly below saturation levels ($\sim 4$ mmol kg$^{-1}$). Thus, mild $Ca^{2+}$ limitation may have masked the impacts of low salinity on calcification. The observed interaction between $[Ca^{2+}]$ and salinity (Table S6) on calcification supports this and as discussed in Sect. 4.2, further experiments at saturated calcium concentrations ($> 4$ mmol kg$^{-1}$) may reveal significant impacts of salinity on calcification in line with the bicarbonate experiment (Fig. 5a). The sampled experimental population at Ahrenshoop is also known to have high genetic diversity (Stuckas et al., 2017) and differences in the genetic composition between juvenile cohorts in both laboratory experiments in this study may explain the better performance at salinity 6 in the calcium experiment compared to the bicarbonate experiment. This coupled with slightly smaller animals in the bicarbonate experiment may be responsible for lower calcification rates at a salinity of 6 and higher mortality rates in the bicarbonate ion experiment. Future work should investigate how population specific changes in genetics and physiological adaptation impacts sensitivity of calcifying mussels to future desalination in the Baltic Sea.

The metabolic costs associated with osmotic stress at low salinities result from cellular mechanisms of volume regulation during salinity changes (Neufeld et al., 1996). Intracellular organic osmolytes (primarily free amino acids and quaternary ammonium compounds) are excreted from cells or mobilised from endogenous protein reserves during hypo- and hyper-salinity exposure, respectively and associated with an energetic cost (Hawkins and Hilbish, 1992). Marine bivalves become iso-osmotic with surrounding seawater once long term ($> 2$ weeks) acclimation to low salinity is complete and subsequently the energetic costs of protein excretion/mobilisation should not persist (Willmer, 1978; Neufeld et al., 1996). Both laboratory experiments in this study lasted $> 30$ days, long enough to overcome the initial energetic costs of salinity acclimation. Despite both experiments having different durations (70 days and 37 days for the bicarbonate and calcium experiments, respectively), it is unlikely that dissimilar exposure times would result in differential impacts of osmotic stress accumulation over time, as experimental durations were long enough to allow physiological acclimation to experimental salinities. Accordingly, the results here suggest that at salinities of 7 - 16, osmotic stress is not the primary cause of reduced growth in Baltic mussels.

### 4.4 Calcification physiology

Reduced calcification rates under sub-optimal carbonate chemistry conditions in this study may be explained in terms of calcification physiology. Adult bivalve molluscs are poor at regulating the ion composition of their extracellular fluid, thus low seawater $[Ca^{2+}]$ directly translates to low haemolymph $[Ca^{2+}]$ with the same concept applying to seawater $pCO_2$, pH and $[HCO_3^-]$. However extracellular carbonate chemistry is even less favourable for calcification than ambient seawater as production of metabolic $CO_2$ maintains extracellular $pCO_2$ higher (and pH lower) than ambient seawater to aid diffusive excretion of metabolic $CO_2$ (Melzner et al., 2009; Heinemann et al., 2012). Sub-optimal carbonate chemistry in the extracellular fluid will therefore be in direct contact with the mantle epithelia (calcifying tissue) and likely impact energy demanding ion transport processes involved in calcification. Increased costs of ion transport and maintenance of transmembrane ion gradients may be the likely driver of increasing mortality rates at lower pH values in the bicarbonate ion experiment (Fig. S10), compared to no mortality in the calcium ion experiment where pH values were similar to field values across all treatments (Table 1 and 3). At a cellular level, low seawater $Ca^{2+}$ availability may impact calcification by reducing rates of calcium transport across the mantle epithelia by plasma membrane $Ca^{2+}$ ATPases (PMCA's) (Niggli et al., 1982; McConnaughey and Whelan, 1997). Although PMCA activity and gene expression has not been shown to be impacted by salinity in *Mercenaria mercenaria* and *Crassoestrea gigas*, respectively, actual rates of $Ca^{2+}$ transport across the mantle epithelia have been shown to decrease at salinities of 14 compared to 28 in *C. gigas* (Ivanina et

al., 2020; Sillanpää et al., 2020). This suggests that the reduced calcification rates observed in this study at low $[Ca^{2+}]$ may stem from a reduced supply of $Ca^{2+}$ to the calcification site. Conversely, protons $(H^+)$ inhibit calcification in the calcifying space by lowering $\Omega$ and imposing constraints on mineral formation (Waldbusser et al., 2014; Cyronak et al., 2016). Organisms may increase the rate of $H^+$ extrusion from the calcifying space (via the V-type $H^+$-ATPase) but this requires more energy in the form of ATP, increasing calcification costs and potentially resulting in less energy available for other processes eg. protein synthesis (Waldbusser et al., 2013; Tresguerres, 2016). Higher energetic costs of calcification have been documented at low salinities (Sanders et al., 2018) and this may be the underlying mechanisms explaining the reduced calcification rates at low salinities and $[Ca^{2+}]$. Increased costs of calcification may also arise from changes in the organic content of the shell. The energetic cost of shell organic matrix proteins synthesis is $\sim$ 20-fold higher than the costs of $CaCO_3$ precipitation (Palmer, 1992) and subsequently, changes in the proportion of shell organic would have major impacts on the metabolic costs of shell production. Shell organic content was not quantified in this study, however previous studies on Baltic *Mytilus* adults have observed higher organic content of shells in mussels living at low salinity (Telesca et al., 2019). Indeed, if this pattern is also true for juvenile mussels then this may be responsible for increased costs of calcification at low salinities. It remains elusive how the organic content of shells may be affected by long term changes in seawater carbonate chemistry and future work on calcification in the oceans should focus on how changes in shell organic content may influence the mechanisms and energetic costs of calcification.

### 4.5 Future environmental trends

Desalination of the Southwest and Central Baltic Proper is projected over the next century due to increased freshwater supply from river discharge resulting from changes in precipitation patterns (Meier et al., 2006; Gräwe et al., 2013). Depending on the composition of the additional freshwater input, this desalination might also suggest a reduction in $A_T$ and $[Ca^{2+}]$. However, $A_T$ has in fact increased in the Central Baltic Sea by $\sim$ 3.4 µmol $kg^{-1}$ $yr^{-1}$ over the past two decades likely due to changes in precipitation and continental weathering (Müller et al., 2016), which has also increased $[Ca^{2+}]$ and $Ca$-$A_T$ ratios over a similar period (Kremling and Wilhelm, 1997). Although future changes in $A_T$ and $[Ca^{2+}]$ may be unclear at present, the magnitude of predicted desalination, coupled with projected reduction in eutrophication and increasing atmospheric $pCO_2$ points towards a trend of decreasing pH in the Central Baltic Sea over the next century (Gustafsson et al., 2019; Gustafsson and Gustafsson, 2020). Future acidification combined with a potential mismatch in future $S$-$Ca^{2+}$ relationships highlight the importance of utilising ESIR as well as $\Omega$, in predicting vulnerability of marine calcifiers in the Southwest and Central Baltic Sea. In the absence of adaptation, projected environmental change may result in a westward distribution shift of marine mussels towards the Kattegat, and/or replacement of calcifying mussels in the Southwest and Central Baltic by brackish or freshwater calcifying bivalves such as the invasive euryhaline freshwater bivalve, *Dreissena* spp. These changes may have potential knock-on effects for benthic biodiversity and ecosystem function in the Southwest and Central Baltic Sea (Vuorinen et al., 2015).

### 5. Conclusions

The environmental gradients in the Baltic Sea provide an excellent system to investigate the impacts of seawater carbonate chemistry on marine calcifying organisms. The results presented here show that extremely slow calcification rates in the Baltic Sea arise from extended periods of $\Omega_{aragonite} < 1$ and low ESIR undersaturation. These sub-optimal conditions for calcification arise primarily from low seawater $[Ca^{2+}]$ rather than low $[HCO_3^-]$ and low pH. Although high food availability and high $A_T$ near the coast may alleviate these negative impacts at salinities $> 11$, this is likely not possible when $[Ca^{2+}]$ is ultimately limiting calcification $< 4$ mm $kg^{-1}$. Predicted acidification and desalination in the Southwest Baltic is likely to impact Baltic *Mytilus* calcification and distributions, however this will depend on future $[Ca^{2+}]$ trends. Upcoming work investigating the impacts of climate change on Baltic Sea calcifying organisms should a) focus on better understanding future changes in $[Ca^{2+}]$ in the Southwest and Central Baltic Proper and b) further investigate the mechanisms and costs associated with cellular $Ca^{2+}$ transport in bivalves and assess their potential to adapt their calcification machinery to a stressful future Baltic Sea environment.

### Data availability

All raw data is available via the Pangaea database: https://doi.org/10.1594/PANGAEA.925017.

### Author contributions

TS, FM and JT designed the laboratory experiments conducted by TS and JT. TS, FM, GR and JDM implemented the environmental monitoring and TS collected and analysed environmental data. JDM and GR analysed carbonate chemistry samples. TS wrote the manuscript with all authors providing significant contributions throughout.

**Competing interests**
The authors declare no conflict of interest.


**Acknowledgements**
The authors thank Stefan Otto at the IOW for carbonate chemistry analysis of samples and Thomas Stegmann at the Christian Albrechts Universität zu Kiel for $Ca^{2+}$ measurements. The authors also thank the Landesamt für Landwirtschaft, Umwelt und Ländliche Räume (LLUR, Kiel) and the Landesamt für Umwelt, Naturschutz und

Geologie (LUNG, Güstrow) for chl-*a* monitoring data. Additionally, the authors would like to thank Luca Telesca (Cambridge University) for assisting the analysis of environmental data. This research was supported by the Marie Curie ITN network 'CACHE' (Calcium in a Changing Environment), European Union Seventh Framework Programme under grant agreement n° 605051. The position of JDM was funded by BONUS, the joint Baltic Sea research and development programme (Art 185), funded jointly from the European Union's Seventh Programme

for research, technological development and demonstration and from the German Federal Ministry of Education and Research through Grant No. 03F0689A (BONUS PINBAL) and Grant No. 03F0773A (BONUS INTEGRAL).

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

**Figures**


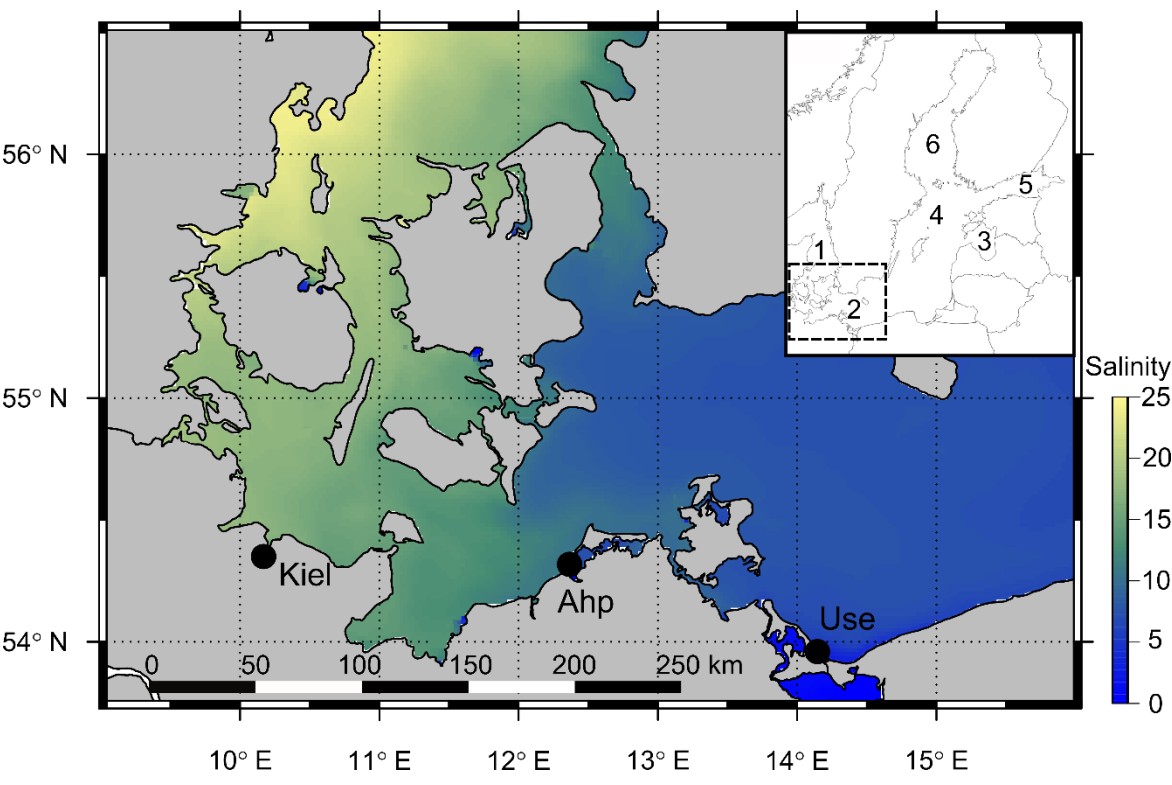

Figure 1: Map of the Southwest and Central Baltic Sea depicting the decreasing salinity gradient from the
Kattegat (West) to the Baltic Proper (East) with the three field sites shown (Kiel, Ahrenshoop and Usedom).
The inset map shows the position of the Southwest Baltic Sea (box in inset) in relation to other regions: 1 –
Kattegat, 2 – Southwest Baltic, 3 – Gulf of Riga, 4 – Central Baltic Proper, 5 – Gulf of Finland and 6 –
Bothnian Sea. Salinity data was taken from open-source monitoring data (EU, Copernicus Marine Services,
2018; details in supplementary material).






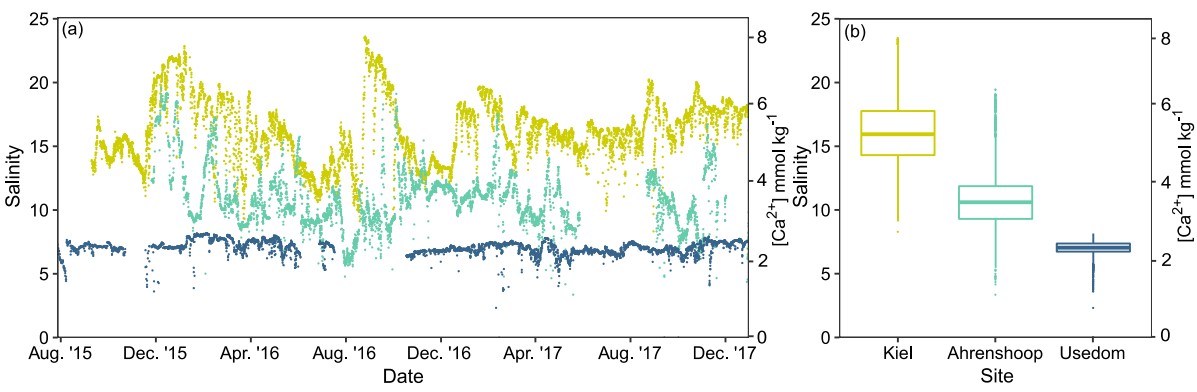

**Figure 2: Field salinity and [Ca$^{2+}$] at the three Baltic Sea sites from Aug. 2015-Dec. 2017 (a). Salinity data is derived from deployed CTD loggers and [Ca$^{2+}$] was calculated from these values (Sect. 2.1). Box plots depict median salinity and interquartile ranges excluding outliers (b).**











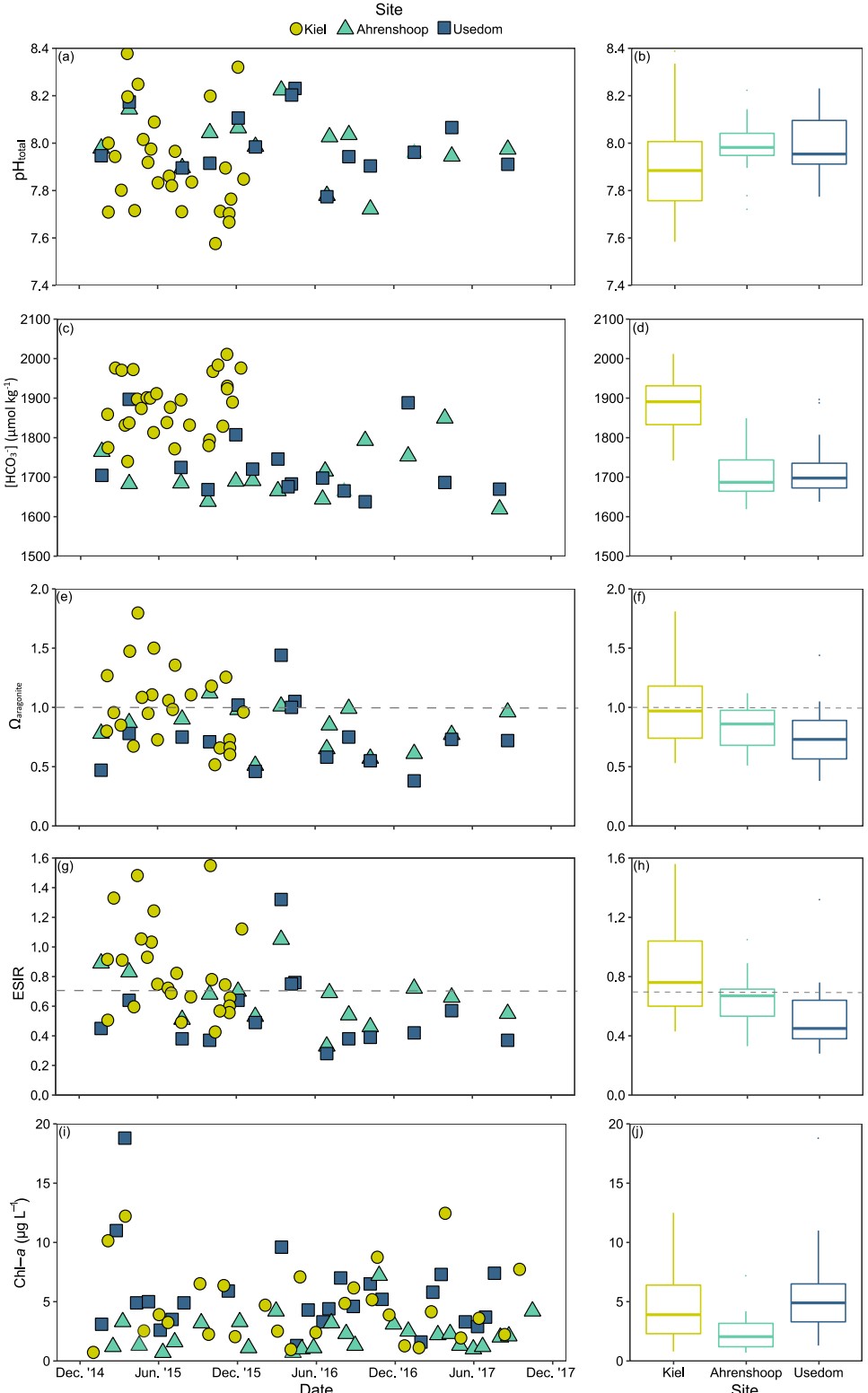

**Figure 3: Environmental monitoring data from Kiel, Ahrenshoop and Usedom. Time series for pH$_{total}$ (-log$_{10}$[H$^+$]) at *in situ* temperatures (a), and associated boxplots (b); [HCO$_3^-$] (c), and boxplots (d); Ω$_{aragonite}$ with the horizontal dashed line depicting the theoretical saturation threshold of Ω$_{aragonite}$ = 1 (e), and boxplots (f); extended substrate-inhibitor ratio ([Ca$^{2+}$][HCO$_3^-$] / [H$^+$]) with the horizontal dashed line depicting the theoretical saturation threshold of ESIR = 0.7 (g), and boxplots (h); Chl-*a* as a proxy for food availability (i), and boxplots (j). Details describing carbonate chemistry calculations are described in Sect. 2.1. Boxplots display median values and interquartile ranges excluding outliers.**



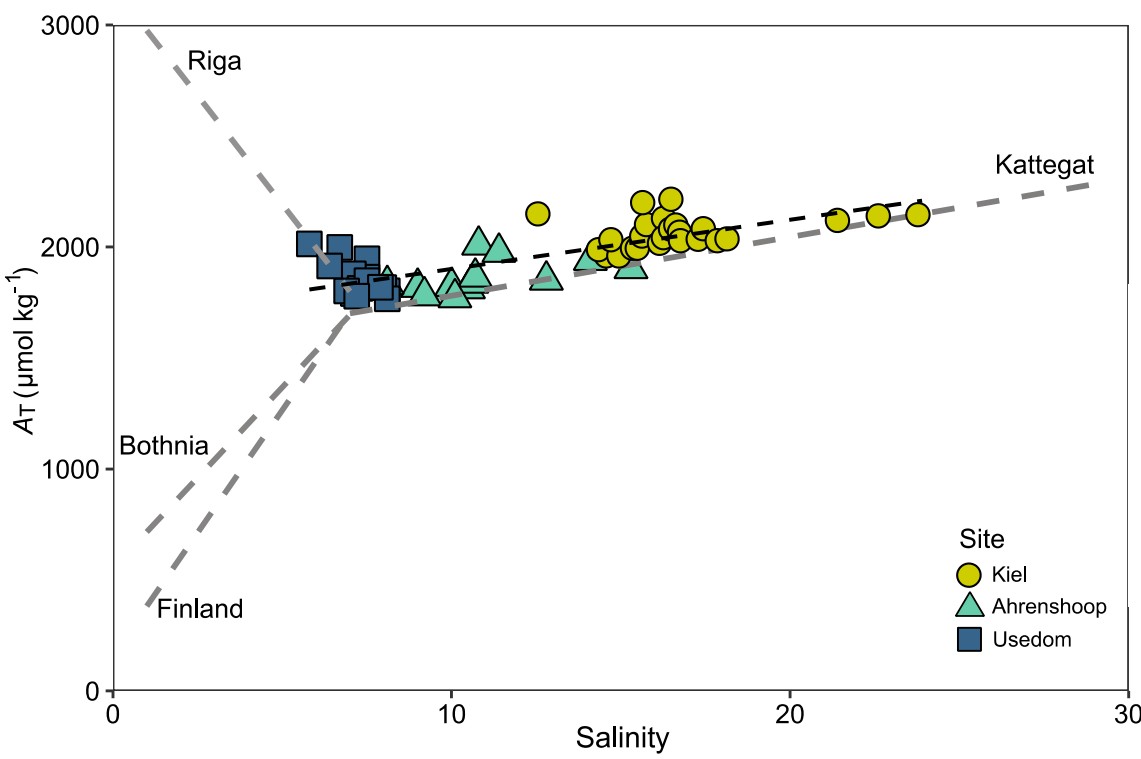

**Figure 4:** $A_T$-S relationship from field monitoring data over the monitoring period for each site showing a linear fit with model parameters given in Table S7. Linear $A_T$-S relationships (grey dashed lines) for the Gulf of Bothnia, Gulf of Finland and the Kattegat are also depicted, using 2D linear model parameters for 2014 from Müller et al., 2016. The linear $A_T$-S relationship for the Gulf of Riga is shown using model parameters for 2008 from Beldowski et al., 2010.

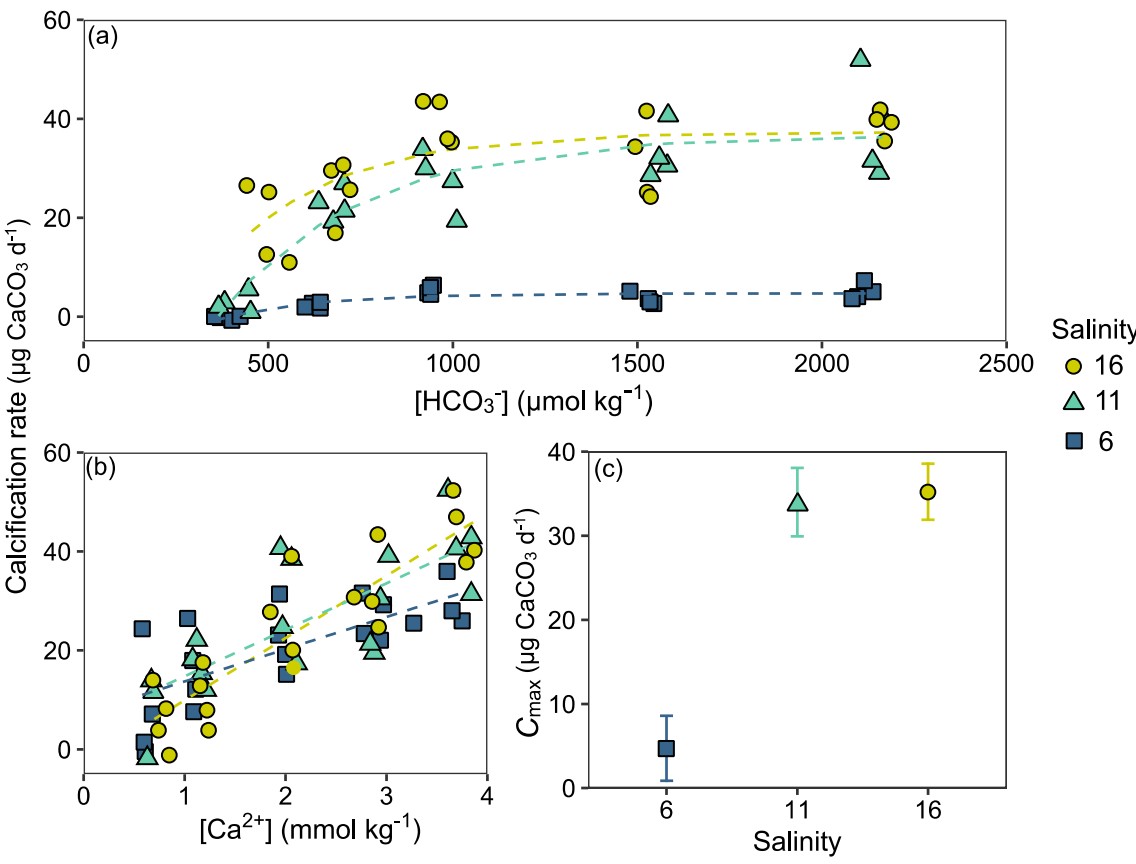

**Figure 5: Individual calcification rates (µg CaCO₃ d⁻¹) in both laboratory experiments. Calcification rates across [HCO₃⁻] in the bicarbonate experiment are shown with negative exponential decay models for each salinity treatment (a). Calcification rates across [Ca²⁺] in the calcium experiment are fitted with linear trendlines for each salinity treatment (b). The model parameter $C_{max}$ (µg CaCO₃ d⁻¹) for each salinity in the bicarbonate experiment is depicted ± 95 % confidence interval (c).**

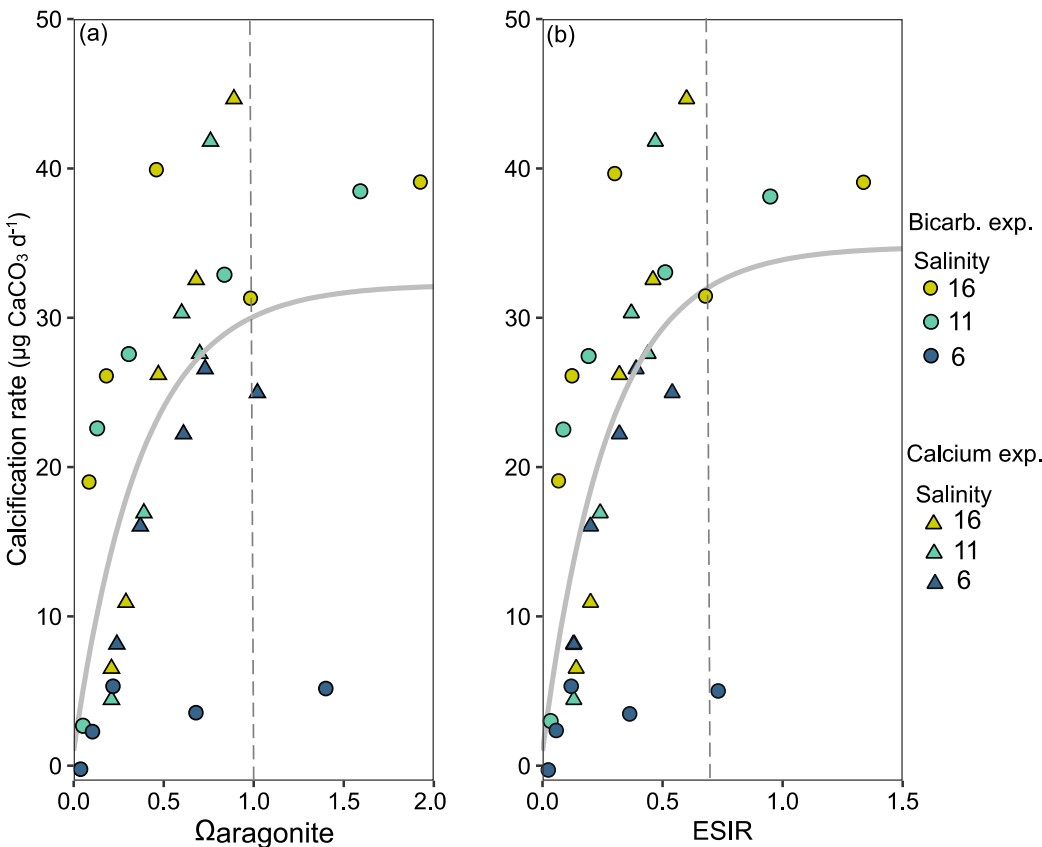

Figure 6: Mean calcification rates in each salinity treatment from both laboratory experiments ([HCO$_3^-$] - circles and [Ca$^{2+}$] - triangles) across calculated $\Omega_{aragonite}$ (a) and ESIR ([Ca$^{2+}$][HCO$_3^-$] / [H$^+$]) (b). Non-linear negative exponential decay models are graphically presented across both laboratory experiments, and vertical dotted lines represent the theoretical saturation thresholds, below which calcification rates are severely impeded ($\Omega_{aragonite}$ = 1, ESIR = 0.7).

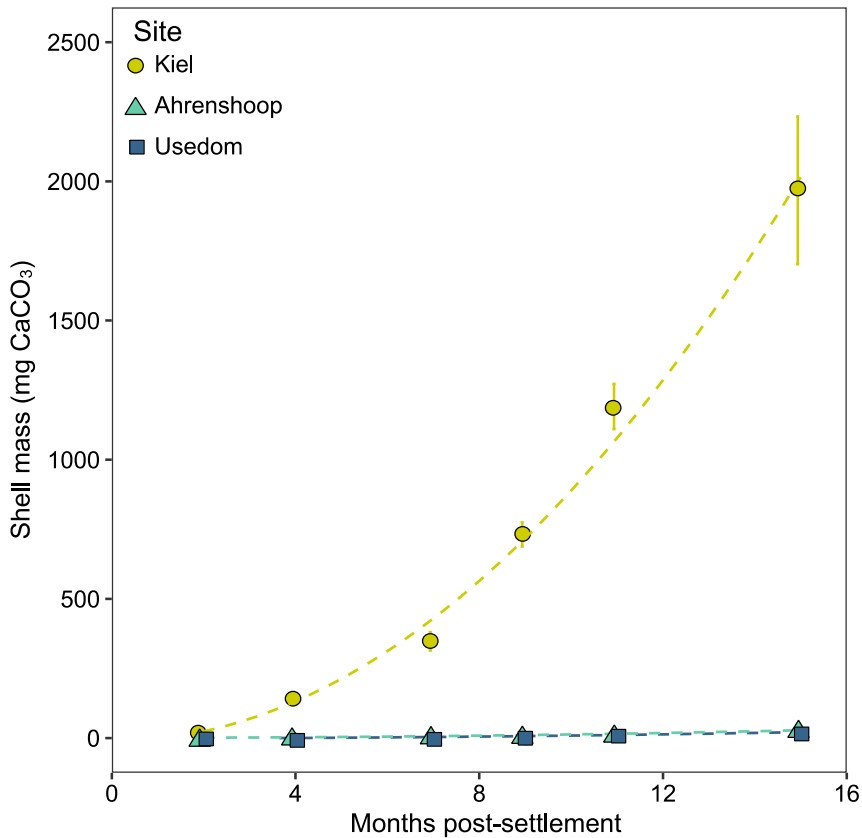

**Figure 7: Field calcification over 15 months at the three monitoring sites in the Baltic Sea (Kiel, Ahrenshoop and Usedom). Shell mass was calculated from shell SL-CaCO$_3$ mass relationships for each population (Fig. S3).**

**Table 1: Summary of environmental conditions and field calcification rates for the experimental monitoring period (2015-2018) at the three Baltic Sea localities (Kiel, Ahrenshoop, Usedom). Salinity and temperature were recorded using mini-CTD's, $pH_{total}$ $C_T$ and $A_T$ were measured in laboratory conditions with $pCO_2$, $[HCO_3^-]$ and $\Omega_{aragonite}$ calculated on CO2Sys_v2.1 programme using $pH_{total}$ and $C_T$ as inputs. $[Ca^{2+}]$ and the ESIR were calculated from carbonate, salinity and carbonate chemistry parameters respectively. The chl-*a* data presented originates from physical monitoring data (Sect. 2.3; Kiel – 28 samples, Ahrenshoop – 25 samples, Usedom – 25 samples). Calcification rates represent the linear slope coefficients of calcification over time (days) at each site with linear model parameters given in Table S10. Mean values are presented ± standard deviation (below).**

| site | salinity | temp. (°C) | Chl-a (mg L$^{-1}$) | $pH_{total}$ | $[Ca^{2+}]$ (mmol kg$^{-1}$) | $pCO_2$ (µatm) | $[HCO_3^-]$ (µmol kg$^{-1}$) | $A_T$ (µmol kg$^{-1}$) | $\Omega_{arag}$ | ESIR | calc. rate (µg CaCO$_3$ d$^{-1}$) |
|---|---|---|---|---|---|---|---|---|---|---|---|
| Kiel | 15.2 ± 3.1 | 11.2 ± 5.1 | 4.68 ± 3.23 | 7.94 ± 0.22 | 4.99 ± 0.91 | 624 ± 292 | 1883 ± 74 | 2071 ± 67 | 1.17 ± 0.51 | 1.05 ± 0.60 | 2202.9 ± 37.64 |
| Ahp | 10.9 ± 2.3 | 10.3 ± 5.8 | 2.25 ± 1.45 | 7.98 ± 0.13 | 3.87 ± 0.75 | 506 ± 172 | 1704 ± 65 | 1858 ± 72 | 0.83 ± 0.19 | 0.65 ± 0.19 | 36.4 ± 53.2 |
| Use | 7.0 ± 0.6 | 10.2 ± 6.4 | 5.52 ± 3.59 | 8.03 ± 0.18 | 2.63 ± 0.17 | 508 ± 186 | 1725 ± 79 | 1856 ± 79 | 0.76 ± 0.28 | 0.55 ± 0.26 | 18.6 ± 53.2 |

**Table 2: Water chemistry in the bicarbonate ion manipulation experiment: Mean parameters for experimental treatments over the course of the experiment. Columns from left to right present: Target and Experimental salinity, $pH_{total}$, partial pressure of carbon dioxide, total alkalinity, total dissolved inorganic carbon, Target and actual bicarbonate ion concentration, carbonate ion concentration, carbon dioxide concentration, calcium ion concentration, aragonite saturation state, substrate inhibitor ratio and extended substrate inhibitor ratio. All carbonate chemistry parameters were calculated using $pH_{NBS}$ and $C_T$ as inputs and $[Ca^{2+}]$ was calculated as described in Sect. 2.1. Mean values are presented for each experimental treatment (n = 4) ± standard deviation with measurements taken immediately before and after water changes.**

| target salinity | salinity | $pH_{total}$ | $pCO_2$ (μatm) | $A_T$ (μmol kg$^{-1}$) | $C_T$ (μmol kg$^{-1}$) | target $[HCO_3^-]$ | $[HCO_3^-]$ (μmol kg$^{-1}$) | $[CO_3^{2-}]$ (μmol kg$^{-1}$) | $CO_2$ (μmol kg$^{-1}$) | $[Ca^{2+}]$ (mmol kg$^{-1}$) | $\Omega_{arag}$ | SIR | ESIR |
|---|---|---|---|---|---|---|---|---|---|---|---|---|---|
| 6.0 | 6.5 | 7.37 | 537.4 | 397.4 | 417.3 | 300 | 392.4 | 2.7 | 22.2 | 2.5 | 0.04 | 0.01 | 0.02 |
| | ± 0.0 | ± 0.03 | ± 19.5 | ± 19.7 | ± 19.0 | | ± 19.0 | ± 0.3 | ± 0.8 | ± 0.0 | ± 0.01 | ± 0.00 | ± 0.00 |
| 6.0 | 6.3 | 7.57 | 549.4 | 643.7 | 660 | 600 | 630.8 | 6.5 | 22.7 | 2.4 | 0.11 | 0.02 | 0.06 |
| | ± 0.0 | ± 0.01 | ± 11.3 | ± 14.1 | ± 13.7 | | ± 13.4 | ± 0.3 | ± 0.5 | ± 0.0 | ± 0.01 | ± 0.00 | ± 0.00 |
| 6.0 | 6.3 | 7.73 | 573.7 | 973.1 | 982.6 | 900 | 944.7 | 14.2 | 23.7 | 2.4 | 0.23 | 0.05 | 0.12 |
| | ± 0.0 | ± 0.02 | ± 29.2 | ± 17.0 | ± 16.7 | | ± 16.1 | ± 0.7 | ± 1.2 | ± 0.0 | ± 0.01 | ± 0.00 | ± 0.01 |
| 6.0 | 6.3 | 8 | 493.6 | 1614.1 | 1592.4 | 1500 | 1529.8 | 42.2 | 20.4 | 2.4 | 0.69 | 0.15 | 0.37 |
| | ± 0.0 | ± 0.01 | ± 10.5 | ± 17.0 | ± 16.0 | | ± 15.1 | ± 1.2 | ± 0.4 | ± 0.0 | ± 0.02 | ± 0.00 | ± 0.01 |
| 6.0. | 6.2 | 8.17 | 462.4 | 2278.6 | 2214.6 | 2100 | 2110.2 | 85.3 | 19.1 | 2.4 | 1.39 | 0.31 | 0.74 |
| | ± 0.0 | ± 0.01 | ± 5.5 | ± 22.0 | ± 20.5 | | ± 19.1 | ± 1.7 | ± 0.2 | ± 0.0 | ± 0.03 | ± 0.01 | ± 0.02 |
| 11.0 | 11.3 | 7.43 | 568.2 | 425.3 | 443.5 | 300 | 416.9 | 3.8 | 22.8 | 4 | 0.06 | 0.01 | 0.04 |
| | ± 0.0 | ± 0.03 | ± 26.0 | ± 22.9 | ± 22.1 | | ± 21.9 | ± 0.4 | ± 1.0 | ± 0.0 | ± 0.01 | ± 0.00 | ± 0.00 |
| 11.0 | 11.2 | 7.5 | 648.7 | 706.7 | 722.3 | 600 | 687.3 | 9 | 26 | 4 | 0.14 | 0.02 | 0.09 |
| | ± 0.0 | ± 0.03 | ± 68.3 | ± 15.8 | ± 16.0 | | ± 15.1 | ± 0.6 | ± 2.7 | ± 0.0 | ± 0.01 | ± 0.00 | ± 0.01 |
| 11.0 | 11.1 | 7.69 | 564 | 1010.5 | 1011 | 900 | 969.1 | 19.2 | 22.6 | 3.9 | 0.31 | 0.05 | 0.19 |
| | ± 0.0 | ± 0.01 | ± 14.7 | ± 26.8 | ± 25.9 | | ± 24.9 | ± 1.0 | ± 0.6 | ± 0.0 | ± 0.02 | ± 0.00 | ± 0.01 |
| 11.0 | 11.1 | 7.91 | 539.4 | 1671.2 | 1637.1 | 1500 | 1563.2 | 52.2 | 21.7 | 3.9 | 0.84 | 0.13 | 0.51 |
| | ± 0.0 | ± 0.01 | ± 11.4 | ± 42.9 | ± 40.3 | | ± 37.8 | ± 2.6 | ± 0.5 | ± 0.0 | ± 0.04 | ± 0.01 | ± 0.03 |
| 11.0 | 11.4 | 8.06 | 533.5 | 2330.2 | 2252.2 | 2100 | 2133 | 97.8 | 21.4 | 3.9 | 1.58 | 0.24 | 0.95 |
| | ± 0.0 | ± 0.01 | ± 8.5 | ± 41.5 | ± 38.1 | | ± 35.1 | ± 3.4 | ± 0.3 | ± 0.0 | ± 0.06 | ± 0.01 | ± 0.03 |
| 16.0 | 16.5 | 7.35 | 588.6 | 522.8 | 536.7 | 300 | 507.6 | 6.1 | 23 | 5.7 | 0.1 | 0.01 | 0.07 |
| | ± 0.0 | ± 0.02 | ± 22.5 | ± 35.6 | ± 34.9 | | ± 33.8 | ± 0.8 | ± 0.9 | ± 0.0 | ± 0.01 | ± 0.00 | ± 0.01 |
| 16.0 | 16.4 | 7.52 | 556.8 | 731.4 | 736.5 | 600 | 702.9 | 11.9 | 21.8 | 5.7 | 0.19 | 0.02 | 0.13 |
| | ± 0.0 | ± 0.01 | ± 9.7 | ± 28.5 | ± 27.8 | | ± 26.7 | ± 0.8 | ± 0.4 | ± 0.0 | ± 0.01 | ± 0.00 | ± 0.01 |
| 16.0 | 16.3 | 7.7 | 523.6 | 1084.8 | 1070.8 | 900 | 1022.5 | 27.9 | 20.5 | 5.6 | 0.45 | 0.06 | 0.31 |
| | ± 0.0 | ± 0.03 | ± 4.9 | ± 70.5 | ± 66.6 | | ± 62.9 | ± 3.8 | ± 0.2 | ± 0.0 | ± 0.06 | ± 0.01 | ± 0.04 |
| 16.0 | 16.3 | 7.87 | 547.3 | 1707 | 1658.9 | 1500 | 1575.6 | 61.9 | 21.4 | 5.6 | 0.99 | 0.12 | 0.69 |
| | ± 0.0 | ± 0.02 | ± 8.2 | ± 85.4 | ± 79.7 | | ± 73.9 | ± 5.9 | ± 0.3 | ± 0.0 | ± 0.09 | ± 0.01 | ± 0.07 |
| 16.0 | 16.2 | 8.03 | 555.3 | 2484.6 | 2377.7 | 2100 | 2235.8 | 120.2 | 21.7 | 5.6 | 1.93 | 0.24 | 1.34 |
| | ± 0.0 | ± 0.01 | ± 12.5 | ± 75.4 | ± 69.7 | | ± 63.8 | ± 6.5 | ± 0.5 | ± 0.0 | ± 0.10 | ± 0.01 | ± 0.07 |

**Table 3: Water chemistry in the calcium ion manipulation experiment: Mean parameters for experimental treatments over the course of the experiment. Columns from left to right present: Target and Experimental salinity, pHtotal, partial pressure of carbon dioxide, total alkalinity, total dissolved inorganic carbon, bicarbonate ion concentration, carbonate ion concentration, carbon dioxide concentration, target and experimental calcium ion concentrations, aragonite saturation state, substrate inhibitor ratio and extended substrate inhibitor ratio. All carbonate chemistry parameters were calculated using $pH_{NBS}$ and $C_T$ as inputs. Mean values are presented for each experimental treatment (n = 4) ± standard deviation with measurements taken immediately before and after water changes.**

| target salinity | salinity | $pH_{total}$ | $pCO_2$ ($\mu$atm) | $A_T$ ($\mu$mol kg$^{-1}$) | $C_T$ ($\mu$mol kg$^{-1}$) | [HCO$_3^-$] ($\mu$mol kg$^{-1}$) | [CO$_3^{2-}$] ($\mu$mol kg$^{-1}$) | CO$_2$ ($\mu$mol kg$^{-1}$) | target [Ca$^{2+}$] | [Ca$^{2+}$] (mmol kg$^{-1}$) | $\Omega_{arag}$ | SIR | ESIR |
|---|---|---|---|---|---|---|---|---|---|---|---|---|---|
| 6.0 | 6 | 7.99 | 651.2 | 1972.5 | 1927.2 | 1835.6 | 64.3 | 27.9 | 0.5 | 0.7 | 0.24 | 0.2 | 0.13 |
|  | ± 0.1 | ± 0.12 | ± 8.3 | ± 18.4 | ± 21.5 | ± 28.2 | ± 15.0 | ± 8.3 |  | ± 0.0 | ± 0.06 | ± 0.05 | ± 0.03 |
| 6.0 | 6 | 7.94 | 716 | 1961.9 | 1927.2 | 1839.9 | 57.3 | 30.7 | 1.0 | 1.1 | 0.37 | 0.18 | 0.2 |
|  | ± 0.1 | ± 0.11 | ± 8.4 | ± 20.7 | ± 21.5 | ± 24.8 | ± 12.7 | ± 8.4 |  | ± 0.0 | ± 0.08 | ± 0.04 | ± 0.04 |
| 6.0 | 6.1 | 7.9 | 796.9 | 1954.4 | 1927.2 | 1840.2 | 53.6 | 34.1 | 2.0 | 2 | 0.61 | 0.16 | 0.32 |
|  | ± 0.1 | ± 0.12 | ± 9.0 | ± 18.4 | ± 21.5 | ± 27.3 | ± 14.7 | ± 9.0 |  | ± 0.0 | ± 0.17 | ± 0.04 | ± 0.09 |
| 6.0 | 6.2 | 7.85 | 847.9 | 1942.4 | 1927.2 | 1847.1 | 44.7 | 36.2 | 3.0 | 2.8 | 0.73 | 0.14 | 0.39 |
|  | ± 0.2 | ± 0.08 | ± 6.5 | ± 26.3 | ± 21.5 | ± 19.5 | ± 8.3 | ± 6.5 |  | ± 0.0 | ± 0.13 | ± 0.03 | ± 0.07 |
| 6.0 | 6.3 | 7.88 | 799.1 | 1948.4 | 1927.2 | 1845.7 | 48.1 | 34.1 | 4.0 | 3.7 | 1.02 | 0.15 | 0.54 |
|  | ± 0.1 | ± 0.09 | ± 6.3 | ± 28.0 | ± 21.5 | ± 19.4 | ± 9.7 | ± 6.3 |  | ± 0.1 | ± 0.21 | ± 0.03 | ± 0.11 |
| 11.0 | 10.6 | 7.95 | 645.2 | 1994.5 | 1936 | 1839.4 | 70.3 | 26.3 | 0.5 | 0.7 | 0.21 | 0.18 | 0.13 |
|  | ± 0.2 | ± 0.10 | ± 5.9 | ± 40.5 | ± 18.5 | ± 11.7 | ± 15.3 | ± 5.9 |  | ± 0.0 | ± 0.05 | ± 0.04 | ± 0.03 |
| 11.0 | 10.6 | 8.01 | 565.2 | 2010.2 | 1936 | 1832.1 | 80.8 | 23.1 | 1.0 | 1.2 | 0.39 | 0.2 | 0.24 |
|  | ± 0.2 | ± 0.10 | ± 4.5 | ± 43.5 | ± 18.5 | ± 17.6 | ± 22.0 | ± 4.5 |  | ± 0.0 | ± 0.11 | ± 0.05 | ± 0.06 |
| 11.0 | 10.7 | 7.97 | 601.7 | 1999.7 | 1936 | 1838.5 | 73 | 24.5 | 2.0 | 2 | 0.6 | 0.18 | 0.37 |
|  | ± 0.3 | ± 0.09 | ± 4.4 | ± 38.2 | ± 18.5 | ± 13.7 | ± 15.6 | ± 4.4 |  | ± 0.0 | ± 0.13 | ± 0.04 | ± 0.08 |
| 11.0 | 10.8 | 7.89 | 707.7 | 1978.1 | 1936 | 1848.8 | 58.4 | 28.8 | 3.0 | 3 | 0.7 | 0.15 | 0.44 |
|  | ± 0.3 | ± 0.05 | ± 3.5 | ± 29.4 | ± 18.5 | ± 15.0 | ± 7.2 | ± 3.5 |  | ± 0.0 | ± 0.09 | ± 0.02 | ± 0.06 |
| 11.0 | 10.8 | 7.81 | 839.6 | 1961.2 | 1936 | 1853 | 48.8 | 34.2 | 4.0 | 3.9 | 0.76 | 0.12 | 0.47 |
|  | ± 0.3 | ± 0.04 | ± 3.0 | ± 25.0 | ± 18.5 | ± 17.0 | ± 5.7 | ± 3.0 |  | ± 0.1 | ± 0.09 | ± 0.01 | ± 0.05 |
| 16.0 | 15.1 | 7.95 | 584.6 | 2020.3 | 1940.8 | 1836.8 | 80.7 | 22.8 | 0.5 | 0.8 | 0.2 | 0.17 | 0.14 |
|  | ± 0.2 | ± 0.07 | ± 3.2 | ± 50.5 | ± 32.2 | ± 23.5 | ± 13.2 | ± 3.2 |  | ± 0.0 | ± 0.0 | ± 0.03 | ± 0.02 |
| 16.0 | 15.1 | 7.94 | 599.1 | 2015.2 | 1940.8 | 1839.9 | 77 | 23.3 | 1.0 | 1.2 | 0.3 | 0.16 | 0.2 |
|  | ± 0.2 | ± 0.05 | ± 2.6 | ± 46.4 | ± 32.2 | ± 25.2 | ± 9.6 | ± 2.6 |  | ± 0.0 | ± 0.0 | ± 0.02 | ± 0.03 |
| 16.0 | 15.3 | 7.92 | 614.1 | 2011.5 | 1940.8 | 1842 | 74.3 | 23.8 | 2.0 | 2 | 0.5 | 0.16 | 0.32 |
|  | ± 0.2 | ± 0.04 | ± 1.9 | ± 43.3 | ± 32.2 | ± 26.5 | ± 7.7 | ± 1.9 |  | ± 0.0 | ± 0.0 | ± 0.02 | ± 0.03 |
| 16.0 | 15.4 | 7.9 | 646.8 | 2007.1 | 1940.8 | 1843.3 | 71.8 | 25.1 | 3.0 | 3 | 0.7 | 0.15 | 0.46 |
|  | ± 0.2 | ± 0.05 | ± 2.6 | ± 45.4 | ± 32.2 | ± 25.9 | ± 9.1 | ± 2.6 |  | ± 0.1 | ± 0.1 | ± 0.02 | ± 0.06 |
| 16.0 | 15.4 | 7.91 | 647.5 | 2010.7 | 1940.8 | 1840.2 | 74.8 | 25.2 | 4.0 | 3.8 | 0.9 | 0.16 | 0.6 |
|  | ± 0.2 | ± 0.07 | ± 3.9 | ± 52.0 | ± 32.2 | ± 23.6 | ± 13.4 | ± 3.9 |  | ± 0.0 | ± 0.2 | ± 0.03 | ± 0.11 |