# Peer review of "Decoupling salinity and carbonate chemistry: Low calcium ion concentration rather than salinity limits calcification in Baltic Sea mussels"

_Biogeosciences, 2020_

## Referee Comment (RC1) · Anonymous Referee #1 · 16 Dec 2020

I would like to congratulate the authors for this complete study that incorporates both field and laboratory experiments. The study, in general, is well-written and does not show important methodological failures. However, I have some specific comments and doubts that I would like authors could respond to.

The information provided in the introduction is sufficient to understand the necessity to perform this research. However, I recommend the authors to try to re-order the paragraphs, because there are some paragraphs that are totally disconnected from the others making it difficult to follow the storyline. For example, the paragraph starting

at L98, in my opinion, would fit better at the beginning when the authors explain the study system.

Specifically, at L51 and following it would nice that authors explain more about the ecosystem function of the study species. Authors only make a small notification about that, surely there are studies about the ecological importance of this species and the formed-beds along the Baltic Sea.

Also, I would like to know if the authors have information if these ecosystem functions change along the gradient (salinity), and if the abundance of this species is sensitive to the gradient informed. This information is interesting to highlight the effects of environmental changes on the different Baltic sea mussel populations.

L131: what is based on the diet supply used? Is it based on field measurements, previous feeding rates reported. Please, add a reference.

L142: Authors pointed out that they use 1600 animals by experimental replicate. The authors did monitor the oxygen availability in the aquarium. I am worried that this animal density could affect the oxygen supply to the experimental aquarium, or change the pH conditions as a product of mussel respiration. The experimental replicates were bubbled while both experiments lasted?

L150 and following: Why the duration of both experiments was not the same? How authors can avoid the time accumulated effects of living in stressful environments. Even if the authors calculated a rate (by day), it is not comparable. I think that this an important issue to discuss as to compare both experiments as the results can be under or over-estimate. The authors measured the calcification rates at the end of each experiment, right? This was no clear to me.

I understand that due to experimental limitations, the volume of the replicates had to be different. However, the final density (mussels ml-1) is too different to compare between both laboratory experiments. This is an issue to discuss in terms of denso-dependency

potential effects on the results observed.

L190 and following. Suddenly, the authors show that a field experiment was also performed. However, nor the introduction or abstract is pointed out. In my opinion, this is a stronghold of this study. Please, try to incorporate this information in the last paragraph of the introduction, as well as in the abstract.

About the field study, the authors collected the laboratory experimental mussels in Ahreenshoop, however, the authors also performed field experiments in the other two extra sites. I understand the objective of this, but this is not explained in the manuscript.

Authors, in the field experiment, estimated calcification rates from the reported SL-CaCO3 relationship. I understand, that this is a unique relationship developed for a specific mussel population. However, after reading the introduction where authors pointed out that there are important differences along the salinity gradient. So, in my opinion, this relationship should be different among mussel populations. This could have important effects on the results. Indeed, why authors did not use the same methodology of the laboratory experiment, could improve the comparison of results.

Authors, in the laboratory experiment, show how they burned shells in order to eliminate organic matter from the shells in order to provide CaCO3 data and estimate calcification rates. Were there differences in the organic matter among populations? This is so important, as many previous studies have shown how marine calcifying organisms show different organic matter concentrations under different environmental conditions (lab or field). If authors could show this data would be very interesting to understand another potential factor affecting calcification rates. Indeed, shell organic matter (periostracum and inter, intra-crystalline organic matters) has a shell protection function under corrosive environments, but also as a substrate to favor crystallization and biomineralization processes.

How many times Chl-a was measured during the field experiment?
I do not have major comments on the results section.

The discussion section is clear and also identified the major limitations of the study which is appreciated. Results are broadly discussed from many points of view, however as it was pointed out above, I miss a discussion of other potential causes that could determine the results. Indeed, biomineralization processes not only incorporate $CaCO_3$ precipitation but they incorporate the secretion of periostracum/shell organic matter which has an important function on biomineralization. Also, the entire biomineralization process is energetic expensive because of the secretion of these shell organic compounds. It would be nice the authors develop this idea as potential causes of the results observed in order to complete the discussion section. If authors can show shell organic matter by treatment, this could help a lot to understand the results. Indeed, this could be a future research topic to develop. In addition, some methodological limitations of the study (pointed out above) such as denso-dependency are not discussed in the discussion.

FIGURES AND TABLES.

I suggest changing the order of figures, first showing the environmental conditions of field study sites, and then the results of calcification rates.

In table 1, I noticed that there are important differences in pH conditions among experimental treatments, how could affect the calcification rates?

---

## Referee Comment (RC2) · Anonymous Referee #2 · 18 Dec 2020

This is a very nice manuscript from Sanders and collaborators dealing with the impact of seawater chemistry on mussel growth rates. The two experiments have been well designed in order to discriminate first the impact of salinity vs. carbonate chemistry changes and second salinity vs. calcium concentration changes. These experiments have been complemented by a field survey covering over 3 years. Monitoring of seawater physico-chemistry and mussel growth have been performed at 3 sites along a decreasing salinity gradient towards the Central Baltic. The study is well introduced although I agree with Reviewer#1 that paragraph L98-112 should be put up front. The

methods are most of the time well explained and the results properly discussed. I have no doubts that this will be a nice contribution to the Biogeosciences journal. Congratulations to the authors!

That being said, I have a few concerns and questions that I would like the authors to answer:

1) I have to say that I was impressed on how many individuals you could fit in 2 L containers (1600 animals, small but still. . .). Since you did not consider a flow-through system and changed the water "only" 2 to 3 times weekly, I am really wondering how would change carbonate chemistry but also ammonium and oxygen concentrations between two water changes. Table 1 and 2 are not clear to me. Do these tables show the conditions in the experimental plastic aquaria and/or in the stock seawater? If measured in the aquaria, when were the samples taken? Before and/or after water changes? Were your aquaria aerated? I apologize in case I missed that in the text.

2) In Table S2, you report on a >50% mortality during the 70 days bicarbonate experiment, as well as an important range (10-75%) across treatments. Did you check whether you had some relationships between mortality rates and the imposed chemical changes? Did you replace the dead organisms? If not, what would be the effect on the amount of food available for each individual? Table S2 is not clear to me, what are these biomass data? At the start of the experiment? At the end? You mention on L173 that biomass per litre was comparable between the 2 experiments while I can read that it was 13.2 mg/L during the Ca2+ exp and 51.5 mg/L during the HCO3- exp, it does not seem comparable to me.

3) I believe there is one aspect (maybe related to the point above) that should be discussed. During the first experiment (bicarbonate), mussels at salinity 6 did not grow much (maybe 5 microg/d; Fig. 2a). What is the reason why they grew much better during the second experiment (Fig. 2b) even when Ca2+ concentrations are below ambient levels (2.5 mmol/kg), reaching rates of 20 microg/d)? Is it due to the

differences in terms of experimental design?

4) As such, I do not believe that trying to fit any model to all data points (pooled from the two experiments) makes much sense (Fig. S4 and S5, but also Fig. 3). At least for a better view on the data, you should identify the dots depending on the experiment and salinity levels.

5) It seems that you over-determined carbonate chemistry during the field survey by measuring pH, CT and AT. It is not clear to me if AT data showed (i.e. Fig. 7) are the ones measured or derived from pH and CT, maybe to clarify. Finally, how do computed AT and measured AT compare? This could be a nice way to identify DOC contribution no?

---

## Author Comment (AC1) · 9 Feb 2021

Reviewer comment

I would like to congratulate the authors for this complete study that incorporates both field and laboratory experiments. The study, in general, is well-written and does not show important methodological failures. However, I have some specific comments and doubts that I would like authors could respond to. The information provided in the introduction is sufficient to understand the necessity to perform this research. However,

I recommend the authors to try to re-order the paragraphs, because there are some paragraphs that are totally disconnected from the others making it difficult to follow the storyline. For example, the paragraph starting at L98, in my opinion, would fit better at the beginning when the authors explain the study system.

Author response

The authors agree with the reviewer's suggestion. The Introduction section will be restructured to improve the flow of the storyline specifically by bringing forward L98-L112.

Reviewer comment

Specifically, at L51 and following it would nice that authors explain more about the ecosystem function of the study species. Authors only make a small notification about that, surely there are studies about the ecological importance of this species and the formed-beds along the Baltic Sea.

Author response

This is an important point raised by the reviewer. Benthic reef-forming mytilid mussels in the Baltic Sea are prominent ecosystem engineers forming extensive mussel reefs that enhance biodiversity and organic carbon flow through benthic ecosystems in the Baltic (Koivisto & Westerbom 2010; Attard et al. 2020). Baltic mussel reefs are particularly dominant at salinities below 10 practical salinity units being one of the primary habitat-forming organisms in the central and eastern Basins (Westerbom et al., 2002). Baltic Mytilus reefs are particularly important in their contribution to benthic ecosystem services in the Baltic Sea having a stronger impact on ecosystem function than both macroalgae beds and seagrass meadows (Heckwolf et al., 2020). This information regarding the functionality of these species in Baltic Sea ecosystems will be added to the introduction after L51.

Reviewer comment

[Figure]

Also, I would like to know if the authors have information if these ecosystem functions change along the gradient (salinity), and if the abundance of this species is sensitive to the gradient informed. This information is interesting to highlight the effects of environmental changes on the different Baltic sea mussel populations.

Author response

Biodiversity in the central and Eastern Baltic Basins (salinity 5 – 8) is ca. 10 % that of the North Sea (salinity >32). Despite this Baltic filter feeding mussels dominate benthic ecosystems at salinities of 5 – 8, albeit at extremely reduced growth rates and body sizes. Several studies have observed reduced growth rates of Baltic Mytilus at low salinities (Sanders et al., 2018) and decreasing biomass and abundance of mussels with decreasing salinity in the Baltic Sea, both in the field and in model simulations (Westerbom et al, 2002; Westerbom et al. 2019; Liénart et al., 2020). It is therefore expected that desalination will result reduced maximum size, abundance and biomass of Baltic Mytilus which will have negative impacts on the associated ecosystem services provided by Baltic mytilid mussels. Comparisons of carbon flow between the Bothnian Sea (filter feeding mussels present) and the Bothnian Bay (absence of filter feedings mussels) reveals significantly reduced ecosystem function upon the loss of filter feedings bivalves at salinities below 5 (Elmgren & Hill, 1997). This highlights the potential ecological impacts of loss of calcifying, filter feedings bivalves in the Central Baltic Sea with predicted desalination. This information on the reduced size and contribution to ecosystem function shall also be added to the introduction at L54 when the impacts of salinity on mussel growth are introduced. The drastic reduction in ecosystem services upon the complete loss of calcifying bivalves in the Baltic Sea, will also be mentioned in this section.

Reviewer comment

L131: what is based on the diet supply used? Is it based on field measurements, previous feeding rates reported. Please, add a reference.

Author response

This feeding regime was used to ensure saturated feeding conditions of 10 000 phytoplankton cells m-1 (Riisgård et al., 2013) based on Baltic Mytilus clearance rates reaching maximums at or above 6-7000 cells ml-1. This justification will be clarified in L131 and the relevant reference will be added to the manuscript.

Reviewer comment

L142: Authors pointed out that they use 1600 animals by experimental replicate. The authors did monitor the oxygen availability in the aquarium. I am worried that this animal density could affect the oxygen supply to the experimental aquarium, or change the pH conditions as a product of mussel respiration. The experimental replicates were bubbled while both experiments lasted?

Author response

In the $HCO_3^-$ experiment, ambient air was aerated through all aquaria to ensure fully aerated seawater (L148). Oxygen concentration was not measured during experiments; however pH was monitored (a proxy for carbon dioxide partial pressure) 3-4 x weekly in the $HCO_3^-$ experiment to ensure pH did not deviate by more than 0.1 units from target values due to mussel respiration or calcification between water changes. The frequency of water changes was increased to 3 x weekly during the $HCO_3^-$ experiment as AT started to increase through the addition of alkalinity in food culture.

In the $Ca^{2+}$ experiment, stock artificial seawater was equilibrated with atmospheric $pCO_2$ and $pO_2$ before water changes. Experimental aquaria were not actively aerated during the experiment as biomass per ml in experimental aquaria was lower (Table S2) and preliminary monitoring of pH revealed no detectable impact of mussel respiration on seawater pH. pH and ATǍň were still monitored 2-3 x per week (immediately before and after water changes) to ensure respiration or calcification did not cause deviations from target values. Tables 1 and 2 show that $pCO_2$ throughout experiments did not

differ hugely from field pCO2 values (Table 3).

The rational behind the frequency of carbonate chemistry monitoring with relation to respiration and biomass density in experimental aquaria will be clarified in L148 of the methods section.

Reviewer comment

L150 and following: Why the duration of both experiments was not the same? How authors can avoid the time accumulated effects of living in stressful environments. Even if the authors calculated a rate (by day), it is not comparable. I think that this an important issue to discuss as to compare both experiments as the results can be under or overestimate. The authors measured the calcification rates at the end of each experiment, right? This was no clear to me.

Author response

Experimental durations were not identical due to practical limitations during experimentation. Since organisms were slightly larger (older) in the Ca2+ experiment compared to the HCO3- experiment (different times of spawning and sampling cohort of juveniles), a shorter experimental period was still sufficient to allow a significant increase in shell mass over the experimental period and significant differences between treatments. A longer experimental duration in the Ca2+ experiment may indeed result in different results both within the experiment and between experiments, however methodological limitations prevented this from being realised. We believe that in the context of the experiments presented here, short term exposures whether 1 or 2 months, would not result in significantly different effects of carbonate chemistry when both experiments are comparing the same species, sample population and life stage.

Shell mass was estimated at the beginning of the experiment (from the shell length-shell mass relationship in Fig. S1) and then directly measured at the end of both experiments (2 time points) Since the experiments were (relatively) short compared to

the field monitoring (12+ months), a linear increase in shell growth was employed for calculating calcification in both experiments, as opposed to a power curve (eg. the relationship modelled for field calcification rates). This makes the comparison of both experiments possible as calcification rates (increase in $CaCO_3$ mass over time) are derived from the same function for both experiments. Section 2.4 will be expanded to clarify that 2 time points were used to measure calcification and that a linear relationship was used to derive calcification rates for both experiments.

Of course, we cannot disregard the potential effects of differential accumulation of stress effects between both experiments, however we took relevant steps to maximise comparability between both experiments:

1: We used mussels originating from the same population and life stage in both experiments. 2: We utilised the same experimental design in both experiments (5 x 3 treatments with 4 replicates). 3: We measured calcification using identical methods between both experiments. 4: To ensure food was not a limiting factor we maintained saturated food conditions in both experiments with comparable availability of phytoplankton cells per unit biomass during experiments. 5. We conducted regular water changes at sufficient frequencies to ensure seawater chemistry was not deviating significantly from target values due to metabolism and calcification thus ensuring respiratory build-up of $CO_2$ or depletion of dissolved inorganic carbon/calcium did not impact calcification rates. 6. Most importantly, we ensured seawater alkalinity was at natural levels across all experiments and salinity treatments. Most other studies that expose calcifying organisms to desalination experimentally dilute seawater using distilled water which also dilutes seawater alkalinity. At salinities below 10, this results in extremely low alkalinities (< 1000 $\mu$mol kg-1) which may negatively impact calcification synergistically in low salinity treatments.

These main points highlighting comparability between both experiments will be made clearer in L166-177 in the methods section and discussed at the end of section 4.1 in the discussion.

Reviewer comment

I understand that due to experimental limitations, the volume of the replicates had to be different. However, the final density (mussels ml-1) is too different to compare between both laboratory experiments. This is an issue to discuss in terms of denso-dependency potential effects on the results observed.

Author response

The reviewer is right to point out that the number of mussels ml-1 was quite different between both experiments. However, since mussels at the beginning of the Ca2+ experiment had ~4 x the body mass of those at the beginning of the HCO3- experiment (despite both cohorts being in the juvenile life stage), we argue a better metric for comparing both experiments is the total mussel biomass per L. This metric is also more applicable for identifying the metabolic effects on seawater chemistry between both experiments. Mean biomass per L was 13.2 mg l-1 at the beginning of the Ca2+ experiment, and 51.5 mg-1 at the beginning of the HCO3- experiment, thus within the same order of magnitude. By the end of the HCO3- experiment, biomass per litre of seawater ranged from 32.6 – 537.1 mg l-1 between the most extreme treatments. This highlights that the range of values within a single experiment exceeds the differences between both experiments showing that relatively, both experimental approaches achieved comparable biomass per litre. We argue that monitoring of clearance rates during the experiment and ensuring saturated feeding conditions, combined with regular water chemistry monitoring and water changes, largely minimised the denso-dependent effects across treatments and experiments. In the methods section (L166-177) of the manuscript, the measures taken by the authors to minimise denso-dependant impacts between experiments will be discussed specifically focusing on the 6 points mentioned in the previous comment.

Reviewer comment

L190 and following. Suddenly, the authors show that a field experiment was also performed. However, nor the introduction or abstract is pointed out. In my opinion, this is a stronghold of this study. Please, try to incorporate this information in the last paragraph of the introduction, as well as in the abstract.

Author response

The reviewer makes a good point here about a strength of the study which could be better communicated. The rational underlying the field experiments will be included in the introduction, specifically by bringing forward L98-112 to the start of the introduction. The methods section will be reordered, starting with the field monitoring methodologies of both abiotic conditions and field calcification rates.

Reviewer comment

About the field study, the authors collected the laboratory experimental mussels in Ahreenshoop, however, the authors also performed field experiments in the other two extra sites. I understand the objective of this, but this is not explained in the manuscript.

Author response

The 3 field monitoring sites were chosen to reflect natural populations living at the 3 experimental salinities (6, 11 and 16). Whereas the experimental population from Ahrenshoop was chosen based on this population being located in the genetic transition zone between Baltic Mytilus edulis and Baltic Mytilus trossulus (Stuckas et al., 2017), allowing the ability to investigate the impacts of desalination but also antagonistic effects of increased salinity. This will be clarified in methods section 2.5-2.7 which will be brought to the beginning of the methods section where it will introduce the natural systems being studied more clearly.

Reviewer comment

Authors, in the field experiment, estimated calcification rates from the reported SLCaCO3 relationship. I understand, that this is a unique relationship developed for a specific mussel population. However, after reading the introduction where authors pointed out that there are important differences along the salinity gradient. So, in my opinion, this relationship should be different among mussel populations. This could have important effects on the results. Indeed, why authors did not use the same methodology of the laboratory experiment, could improve the comparison of results.

Author response

Unique SL-CaCO3 relationships were developed for all 3 mussel populations (Sanders et al., 2018) and these relationships are presented in the supplementary material (Fig. S1). The same SL-CaCO3 relationship for the mussel population from Ahrenshoop was used in both laboratory experiments for estimating initial CaCO3 mass (before experimental exposures), as well as the field monitoring study on the Ahrenshoop population. The Kiel and Usedom populations had their own, population specific SL-CaCOǍ3Ǎ relationships used to calculate shell mass in the field study, exactly as pointed out by the reviewer. Throughout the field monitoring study, population specific SL-CaCO3 relationships were used to calculate shell mass from shell lengths, as the number of mussels being collected was so large. These population specific relationships are mentioned in L229, but will be made clearer in this section.

Reviewer comment

Authors, in the laboratory experiment, show how they burned shells in order to eliminate organic matter from the shells in order to provide CaCO3 data and estimate calcification rates. Were there differences in the organic matter among populations? This is so important, as many previous studies have shown how marine calcifying organisms show different organic matter concentrations under different environmental conditions (lab or field). If authors could show this data would be very interesting to understand another potential factor affecting calcification rates. Indeed, shell organic matter (periostracum and inter, intra-crystalline organic matters) has a shell protection function under corrosive environments, but also as a substrate to favor crystallization and biomineralization processes.

Author response

The reviewer makes an excellent and very important point here. In the laboratory experiments, initial CaCO3Âň mass was calculated using a population specific SL-CaCO3 relationship, whereas CaCO3 mass in each treatment was measured using the muffle furnace method described in the methods section. Subsequently, there is no initial measure of shell organic (periostracum or shell matrix proteins) to calculate changes in shell organic content during the course of both experiments. Unpublished work, separate from the data presented in this study for the 3 different populations revealed shell organic content (derived from ashing of shells) to be ∼ 5-10 %, within the range expected for marine molluscs (Palmer 1983; Thomsen et al., 2013), however these data did not reveal any differences between populations and values were variable possibly due to shell biofouling. Previous work by authors and colleagues in the Baltic Sea have shown that organic content of larger adult marine mytilid shells is higher at lower salinities, however the impact of salinity is minor compared with the effects of food availability and shell length on shell organic content (Telesca et al., 2019). However, little is known about how shell organic may differ between populations and experimental treatments in juvenile mussels. Understanding how shell organic content may be modulated in Baltic mussel shells in light of predicted climate change is an important point for understanding the fate of calcifying Baltic mussels. The potential increase in energetic costs of shell production related to higher organic content of shells will be mentioned in L415. Additionally, the role of the periostracum in defending the shell from dissolution will be mentioned in section 4.3 of the discussion.

Reviewer comment

How many times Chl-a was measured during the field experiment?

Author response

The number of Chl-a measurements are as follows: Usedom: 25 data points; Ahrenshoop: 25 data points; Kiel: 28 data points. All Chl-a monitoring occurred from January

2015 – December 2017. This information will be added to the header of Table 3 in the manuscript.

Reviewer comment

I do not have major comments on the results section.

The discussion section is clear and also identified the major limitations of the study which is appreciated. Results are broadly discussed from many points of view, however as it was pointed out above, I miss a discussion of other potential causes that could determine the results. Indeed, biomineralization processes not only incorporate CaCO3 precipitation but they incorporate the secretion of periostracum/shell organic matter which has an important function on biomineralization. Also, the entire biomineralization process is energetic expensive because of the secretion of these shell organic compounds. It would be nice the authors develop this idea as potential causes of the results observed in order to complete the discussion section. If authors can show shell organic matter by treatment, this could help a lot to understand the results. Indeed, this could be a future research topic to develop. In addition, some methodological limitations of the study (pointed out above) such as denso-dependency are not discussed in the discussion.

Author response

The authors agree that including a discussion on shell organic content and energetics would strengthen the study and identify potential new avenues for research, as mentioned above. The authors also agree that density-dependant effects of animals in experimental aquaria are important to consider. However as mentioned, we took multiple steps to minimise the impacts of these effects and we strongly believe findings are comparable between both experiments due to experimental animals arising from the same population and at the same life stage, similar mean biomass per litre, regular water changes to prevent deviation in carbonate chemistry resulting from metabolic or growth impacts, and saturated feeding conditions removing any effect of differential

energy intake between experiments (See response to previous reviewer comment). As mentioned above, we will discuss how we maximised comparability between experiments and how we overcame the slight differences in mean biomass density. This information will be discussed at the end of section 4.1.

Reviewer comment

FIGURES AND TABLES.

I suggest changing the order of figures, first showing the environmental conditions of field study sites, and then the results of calcification rates.

Author response

The authors agree this would improve the flow and presentation of the study. As such this will be implemented in the revised manuscript. This will also reflect adjustments in the ordering of the introduction, methodologies and results sections by starting with investigating field conditions and then moving on to the laboratory methods and results.

Reviewer comment

In table 1, I noticed that there are important differences in pH conditions among experimental treatments, how could affect the calcification rates?

Author response

The authors agree with the reviewer, and we have considered the impacts of pH on calcification rates between experimental treatments through inclusion of [H+] and [CO32-] in both ESIR and Ωaragonite. The effects of pH have also been discussed (L339-L354) with emphasis on the co-linearity of pH, [H+] and [CO32-] and the inability to individually isolate the impacts of each parameter.

References

Attard, K. M., Rodil, I. F., Berg, P., Mogg, A. O. M., Westerbom, M., Norkko, A.,

Gludd, R. N.: Metabolism of a subtidal rocky mussel reef in a high-temperate setting: pathways of organic C flow, Mar. Ecol. Prog. Ser., 645, 41-54, DOI: https://doi.org/10.3354/meps13372, 2020.

Elmgren, R., Hill, C.: Ecosystem function at low biodiversity – the Baltic example, In: Ormond, R., Gage, J., Grassle, J. F., (eds), Marine Biodiversity: patterns and processes, Cambridge University Press, Cambridge, 139-336, 1997.

Heckwolf, M. J., Peterson, A., Jänes, H., Horne, P., Künne, J., Liversage, K., Sajeva, M., Reusch, T. B. H., Kotta, J.: From ecosystems to socio-economic benefits: A systematic review of coastal ecosystem services in the Baltic Sea, Sci. Total Environ., 755, 142565, DOI: https://doi.org/10.1016/j.scitotenv.2020.142565, 2021.

Koivisto, M. E., Westerbom, M.: Habitat structure and complexity as determinants of biodiversity in blue mussel beds on sublittoral rocky shores, Mar. Biol., 157, 1463-1474, DOI: 10.1007/s00227-010-1421-9, 2010.

Liénart, C, Garbaras, A., Qvarfordt, S., Sysoev, A. Ö., Höglander, H., Walve, J., Schagerström, E., Eklöf, J., Karlson, A. M. L.: Long-term changes in trophic ecology of blue mussels in a rapidly changing ecosystem, Limnol. Oceanogr., 9999, 1-17, DOI: https://doi.org/10.1002/lno.11633, 2020.

Palmer, A. R.: Relative cost of producing skeletal organic matrix versus calcification: evidence from marine gastropods, Mar. Biol., 75, 287-292, DOI: https://doi.org/10.1007/BF00406014, 1983.

Riisgård, H. U., Pleissner, D., Lundgreen, K., LArse, P. S.: Growth of mussels Mytlus edulis, at algal (Rhodomonas salina) concentrations below and above saturation levels for reduced filtration rate, Mar. Biol. Res., 9, 1005-1017, DOI: http://dx.doi.org/10.1080/17451000.2012.742549, 2013.

Sanders, T., Schmittmann, L., Nascimento-Schulze, J. and Melzner, F.: High calcification costs limit mussel growth at low salinity, Front. Mar. Sci., 5, 352,

https://doi.org/10.3389/fmars.2018.00352, 2018.

Stuckas, H., Knöbel, L., Schade, H., Breusing, C., Hinrichsen, H. H., Bartel, M., Langguth, C., Melzner, F.: Combining hydrodynamic modelling with genetics: can passive larval drift shape the genetic structure of Baltic Mytilus populations? Mol. Ecol. 26, 2765-2782, https://doi.org/10.1111/mec.14075, 2017.

Telesca, L., Peck, L. S., Sanders, T., Thyrring, J., Sejr, M. K., Harper, E. M.: Biomineralization plasticity and environmental heterogeneity predict geographical resilience patterns of foundation species to future change, Glob. Chang. Biol., 25, 4179-4193, DOI: 10.1111/gcb.14758, 2019.

Thomsen, J., Casties, I., Pansch, C., Körtzinger, A. and Melzner, F.: Food availability outweighs ocean acidification effects in juvenile Mytilus edulis: laboratory and field experiments, Glob. Chang. Biol., 19, 1017-1027, https://doi.org/10.1111/gcb.12109, 2013.

Westerbom, M., Kilpi, M., Mustonen, O.: Blue mussels, Mytilus edulis, at the edge of the range: population structure, growth and biomass along a salinity gradient in the north-eastern Baltic Sea, Mar. Biol., 140, 991-999, DOI: 10.1007/s00227-001-0765-6, 2002.

Westerbom, M., Mustonen, O., Jaatinen, K., Kilpi, M., Norkko, A.: Population dynamics at the range margin: Implications of climate change on sublittoral blue mussels (Mytilus trossulus), Front. Mar. Sci., 6, 292, DOI: https://doi.org/10.3389/fmars.2019.00292, 2019.

---

## Author Comment (AC2) · 9 Feb 2021

Reviewer comment

This is a very nice manuscript from Sanders and collaborators dealing with the impact of seawater chemistry on mussel growth rates. The two experiments have been well designed in order to discriminate first the impact of salinity vs. carbonate chemistry changes and second salinity vs. calcium concentration changes. These experiments have been complemented by a field survey covering over 3 years. Monitoring of seawater physico-chemistry and mussel growth have been performed at 3 sites along a decreasing salinity gradient towards the Central Baltic. The study is well introduced although I agree with Reviewer#1 that paragraph L98-112 should be put up front. The emethods are most of the time well explained and the results properly discussed. I have no doubts that this will be a nice contribution to the Biogeosciences journal. Congratulations to the authors!

That being said, I have a few concerns and questions that I would like the authors to answer: 1) I have to say that I was impressed on how many individuals you could fit in 2 L containers (1600 animals, small but still. . .). Since you did not consider a flow-through system and changed the water "only" 2 to 3 times weekly, I am really wondering how would change carbonate chemistry but also ammonium and oxygen concentrations between two water changes. Table 1 and 2 are not clear to me. Do these tables show the conditions in the experimental plastic aquaria and/or in the stock seawater? If measured in the aquaria, when were the samples taken? Before and/or after water changes? Were your aquaria aerated? I apologize in case I missed that in the text.

Author response

The reviewer makes an important point here. From the beginning of the experiment, pH and carbonate chemistry were monitored 3-4 x per week to observe the impacts of metabolism and calcification on experimental seawater conditions. Resultingly, water changes were initiated 2 x weekly at the beginning of the experiment as this was found to be sufficient to prevent significant deviations of more than 0.2 mmol kg-1 $Ca^{2+}$ and 200 $\mu$mol kg-1 $HCO_3$- in seawater chemistry due to biological activity. The depletion of $HCO_3$- and $Ca^{2+}$ due to calcification in both experiments was also partially compensated by the addition of phytoplankton food which was cultured in filtered Kiel fjord water at a salinity of 16 ([$HCO_3$-] = 1883 $\mu$mol kg-1, [$Ca^{2+}$] = 4.99 mmol kg-1). The frequency of water changes was increased to 3 x weekly towards the second half of the experiment because the requirement to add more phytoplankton food resulted in

increasing HCO3 concentrations in treatment aquaria by more than 200 $\mu$mol kg-1. Water changes in the calcium experiment were kept at a frequency of twice weekly as the combination of food addition and calcification did not cause deviations in Ca2+ concentrations by more than 0.2 mmol kg-1 across all treatments.

Tables 1 & 2 present mean values from each experimental treatment (4 x replicate aquaria) over the course of the experiment. Measurements were taken immediately before and after water changes to present the range of experimental water chemistries within treatments (This will be clarified in section 2.2 L148 and L163). In the HCO3-experiment (1600 animals), aquaria were equilibrated with ambient air at atmospheric pO2 and pCO2 equilibrium (L148) to ensure air saturation. In the Ca2+ experiment, 2-3 weekly monitoring of experimental seawater chemistry revealed minimal impacts of biological activity on pH (ie. pHNBS standard errors of less than 0.1 pH units), even though these 60 ml aquaria were not aerated during experiments. Oxygen was not measured, but changes in pH (resulting from CO2 or net acid excretion) was used as a proxy for the impacts of metabolism on seawater chemistry. Values for pCO2 in both experiments (table 1 & 2) revealed laboratory experiments were remarkably similar to field conditions across all 3 sites and salinities, and pH did not vary by more than 0.1 units between water changes across all treatments and experiments. Ammonia excretion was not quantified, however given a conservatively high ammonia excretion rate in Baltic mussels of 20 $\mu$g NH4 per gram dry weight hr-1 (Tedengren & Kautsky 1987), the more biomass dense HCO3- experiment (mean biomass 52 mg dry weight per litre) would have resulted in maximum ammonia concentration of 0.08 mg L-1 immediately prior to a water change after 3 days accumulation. This value of 0.08 mg L-1 is far below the LC50 value for juvenile Mytilus edulis of 0.39 mg NH4 L-1 after 21 days exposure (Kennedy et al., 2017). Subsequently we do not believe there to be any negative impacts of ammonia build-up in either experiments.

The rational behind the frequency of water changes and the monitoring of pH as a proxy for monitoring the impacts of respiration and calcification on seawater chemistry

and the monitoring of [Ca2+] and [HCO3-], will be clarified in methods section 2.2.

Reviewer comment

2) In Table S2, you report on a >50% mortality during the 70 days bicarbonate experiment, as well as an important range (10-75%) across treatments. Did you check whether you had some relationships between mortality rates and the imposed chemical changes? Did you replace the dead organisms? If not, what would be the effect on the amount of food available for each individual? Table S2 is not clear to me, what are these biomass data? At the start of the experiment? At the end? You mention on L173 that biomass per litre was comparable between the 2 experiments while I can read that it was 13.2 mg/L during the Ca2+ exp and 51.5 mg/L during the HCO3- exp, it does not seem comparable to me.

Author response

Mortality rates did exhibit patterns in relation to seawater chemistry. Mortality rates were highest at low pH/[HCO3-] and slightly higher at low salinities (6) compared to higher salinities of 11 and 16 (see attached Figure 1). Table S2 presents the standard deviation of mortality rates across all treatments in the experiment. Dead organisms were not replaced as this would introduce issues related to sizes and differential exposure times of organisms within treatments. Feeding regimes were chosen in such a way to ensure saturated feeding conditions in all treatments (>10 000 phytoplankton cells ml-1). To correct for larger biomasses in certain experimental treatments, feeding frequencies were increased to prevent energy intake from becoming limiting. Clearance rates in each aquaria were monitored (L172) every 2 weeks to ensure sufficient frequencies of feeding as biomass in aquaria increased (growth) or decreased (mortality). This information is presented (L168) but will be expanded for the sake of clarification.

Table S2 presents total biomass per replicate tank as a mean value over the entire experimental period. Both experiments are presented as a comparison, as well as the

range of values within the HCO3- experiment to highlight that biomass and biomass per ml varied by a higher degree within the HCO3$\check{\mathrm{A}}$ň- experiment that between both experiments. The mean values for each experiment (13.2 mg/L and 51.5 mg/L) are therefore within the same order of magnitude. As mentioned in our responses to the comments by Reviewer 1, measures were taken to ensure maximum comparability between both experiments (See author responses to reviewer 1). The header for Table S2 will be clarified to reflect exactly what these data represent.

Reviewer comment

3) I believe there is one aspect (maybe related to the point above) that should be discussed. During the first experiment (bicarbonate), mussels at salinity 6 did not grow much (maybe 5 microg/d; Fig. 2a). What is the reason why they grew much better during the second experiment (Fig. 2b) even when Ca2+ concentrations are below ambient levels (2.5 mmol/kg), reaching rates of 20 microg/d? Is it due to the differences in terms of experimental design?

Author response

This is an interesting point raised by the reviewer. It is true that that calcification rates at a salinity of 6 in the HCO3- experiment are significantly lower than calcification rates at a salinity of 6 in the Ca2+ experiment even at comparable [Ca2+]. A reason for this may be that the animals in the HCO3- experiment were younger/smaller, and therefore more sensitive to adverse changes in seawater carbonate chemistry/pH. This has been suggested to be related to higher calcification rates relative to body mass in larval and juvenile mussels compared to adults (Thomsen et al., 2015). Older, larger juveniles in the Ca2+ experiment may therefore be more resilient to low salinity/pH/[Ca2+], which may also explain the lack of mortality in this experiment. The high genetic diversity observed in the sampled experimental population (Ahrenshoop) could also result in genetic differences between the cohorts which may explain differential tolerances to salinity 6 between both experiments (Stuckas et al., 2017).

Statistical analyses in the Ca2+ experiment do suggest similar impacts of low salinity (6) between both experiments however, as a significant interaction effect between salinity and [Ca2+] may result from lower calcification rates at a salinity of 6 compared to 11 and 16 (Table S6). The differences in calcification rates at a salinity of 6 between both experiments will be discussed in terms of mortality rates and potential genetic differences between cohorts in section 4.3 the revised manuscript.

Reviewer comment

4) As such, I do not believe that trying to fit any model to all data points (pooled from the two experiments) makes much sense (Fig. S4 and S5, but also Fig. 3). At least for a better view on the data, you should identify the dots depending on the experiment and salinity levels.

Author response

The authors agree that fitting one model to data from multiple experiments has its limitations. However, we argue, experiments were performed in such a way to maximise comparability. Calcification rates were comparable between both experiments at high salinities and [HCO3-]/[Ca2+] as well as in the field at Ahrenshoop and Usedom sites, suggesting that both experiments simulated natural environmental conditions equally well.

Fig. 3 depicts both experiments separately (black triangles and red dots), however to visualise both experiments more transparently, each salinity treatment (6, 11 and 16) will also be highlighted in Fig 3 (and Figs S4 and S5) in the revised manuscript.

Reviewer comment

5) It seems that you over-determined carbonate chemistry during the field survey by measuring pH, CT and AT. It is not clear to me if AT data showed (i.e. Fig. 7) are the ones measured or derived from pH and CT, maybe to clarify. Finally, how do computed AT and measured AT compare? This could be a nice way to identify DOC contribution

no?

Author response

AT, CT and pH were all determined from field samples, however only pH and CT were used to calculate other carbonate chemistry parameters due to the potential impacts of dissolved organic matter on AT, as the reviewer mentioned. AT data is not shown in Fig. 7, but rather the subsequent carbonate chemistry parameters calculated from field measurements of pH and CT. The reviewer makes an interesting point in comparing measured AT from field samples, and calculated AT from field CT and pH. However, the potential contribution of DOM towards AT is complex, as this organic alkalinity contribution (Aorg) is not a linear function of DOC, but rather dependent on various parameters such as pH. Previous studies found that Aorg ranged from 22–58 $\mu$mol kg-1, and developed the first mechanistic understanding of how this contribution relates to the amount and nature of DOM, as well as seawater carbonate chemistry (Kuliński et al., 2014). Simply comparing our measured AT and calculated AT would not help us to better understand the Aorg contribution to alkalinity, while a detailed analysis of this contribution is well beyond the scope of this paper. Thus, we prefer not to include the suggested comparison.

Figures

Figure 1: Total mortality rates from an initial abundance of 1600 animals per aquarium for each treatment (N = 4) over the course of the 70-day bicarbonate ion manipulation experiment. Linear fits are shown for each salinity (6, 11 and 16).

References

Kennedy, A. J., Lindsay, J. H., Biedenbach, J. M., Harmon, A. R.: Life stage sensitivity of the marine mussel Mytilus edulis to ammonia, Environ. Toxicol. Chem. 36, 89-95, DOI: 10.1002/etc.3499, 2017.

Kuliński, K., Schneider, B., Hammer, K., Machulik, U. and Schulz-Bull, D.: The influence of dissolved organic matter on the acid–base system of the Baltic Sea, J. Mar. Syst., 132, 106–115, https://doi.org/10.1016/j.jmarsys.2014.01.011, 2014.

Stuckas, H., Knöbel, L., Schade, H., Breusing, C., Hinrichsen, H. H., Bartel, M., Langguth, C., Melzner, F.: Combining hydrodynamic modelling with genetics: can passive larval drift shape the genetic structure of Baltic Mytilus populations? Mol. Ecol. 26, 2765-2782, https://doi.org/10.1111/mec.14075, 2017.

Tedengren, M., Kautsky, N.: Comparative study of the physiology and its probably effect on size in blue mussels (Mytilu edulis L.) from the North Sea and the northern Baltic Proper, Ophelia, 25, 147-155, DOI: 10.1080/00785326.1986.10429746, 1986.

Thomsen, J., Haynert, K., Wegner, K. M. and Melzner, F.: Impact of seawater carbonate chemistry on the calcification of marine bivalves, Biogeosciences, 12, 4209-4220, https://doi.org/10.5194/bg-12-4209-2015, 2015.

**Fig. 1.**

---

## Author Response (AR1)

**Reviewer 1**

I would like to congratulate the authors for this complete study that incorporates both field and laboratory experiments. The study, in general, is well-written and does not show important methodological failures. However, I have some specific comments and doubts that I would like authors could respond to.

The information provided in the introduction is sufficient to understand the necessity to perform this research. However, I recommend the authors to try to re-order the paragraphs, because there are some paragraphs that are totally disconnected from the others making it difficult to follow the storyline. For example, the paragraph starting at L98, in my opinion, would fit better at the beginning when the authors explain the study system.

Author response

**The authors agree with the reviewer's suggestion. The Introduction section has been restructured with the first paragraph focusing on the environmental conditions present in the Baltic Sea, the second paragraph introducing the ecological importance of calcifying mussels and what is currently known about how the environment impacts calcification rates and the following paragraphs introducing the known mechanisms of how salinity and carbonate chemistry impacts calcification. These revisions have been made in L42-126.**

Specifically, at L51 and following it would nice that authors explain more about the ecosystem function of the study species. Authors only make a small notification about that, surely there are studies about the ecological importance of this species and the formed-beds along the Baltic Sea.

Author response

**We have provided more details in the Introduction of the ecological dominance and role calcifying mussels play in benthic Baltic Sea ecosystems. Consideration has been given to the degree of ecosystem services provided by biogenic mussel reefs compared to other biogenic habitats and several ecological functions provided by mussels have also been mentioned (L52-61). Finally, the potential for mussel aquaculture as a means of regional remediation of eutrophication has also been mentioned (L61-63), highlighting the dominant ecological role played by marine calcifying mussels in the Baltic Sea.**

Also, I would like to know if the authors have information if these ecosystem functions change along the gradient (salinity), and if the abundance of this species is sensitive to the gradient informed. This information is interesting to highlight the effects of environmental changes on the different Baltic sea mussel populations.

Author response

**The authors have reworded the introduction to reflect the documented change in growth rates biomass and abundance of Baltic *Mytilus* down to salinities of 5 (L66-69). The functional contribution of Baltic *Mytilus* to ecosystems services has been commented on in relation to the salinity gradient and emphasis has been put on the fact that ecosystem function drops drastically below salinities of 5 when calcifying mussels are no longer present. This revised section of the introduction further highlights the ecological importance of Baltic *Mytilus* and how potential changes in growth rates, biomass and abundances can have large ecological consequences.**

L131: what is based on the diet supply used? Is it based on field measurements, previous feeding rates reported. Please, add a reference.

Author response

**The rational behind the choice of this feeding regime has been clarified in the methods section (L193-194) based on previous studies investigating feeding rates in Baltic *Mytilus*, and the accompanying study has been referenced (Riisgård et al., 2013).**

L142: Authors pointed out that they use 1600 animals by experimental replicate. The authors did monitor the oxygen availability in the aquarium. I am worried that this animal density could affect the oxygen supply to the experimental aquarium, or change the pH conditions as a product of mussel respiration. The experimental replicates were bubbled while both experiments lasted?

Author response

**The methods section has been rewritten to include steps taken by authors to ensure animal densities in both experiments did not impact seawater chemistry through respiration, calcification or food addition. Due to minor differences in initial mean sizes between both cohorts of mussels in either experiment, we aimed to ensure biomass density per aquaria were comparable between both experiments (Table S2; L354-356). The methods section now states that experimental aquaria were aerated in the bicarbonate experiment (L206) but not in the calcium experiment (L239). The frequency of water chemistry monitoring and water changes has been clarified with mention of the maximum acceptable deviation in pH, $[Ca^{2+}]$ and $[HCO_3^-]$ between water changes (L216-218 and L240-241). Water chemistry monitoring revealed mean pH deviated by between 0.04 and 0.12 units between water changes, in line with the pH variation observed at field monitoring sites (Table 1). Changes in experimental pH values before and after water changes (presented in table S2) have been described in the results section (L351-353). Oxygen saturation levels in experimental aquaria were not measured, however due to minor (< 0.1) deviations in pH values between water changes in experimental aquaria we are confident oxygen levels in the bicarbonate ion experiment did not drop significantly due to the high number of animals (See our response to reviewer 2's comment, L245-268 of this document).**

L150 and following: Why the duration of both experiments was not the same? How authors can avoid the time accumulated effects of living in stressful environments. Even if the authors calculated a rate (by day), it is not comparable. I think that this an important issue to discuss as to compare both experiments as the results can be under or overestimate. The authors measured the calcification rates at the end of each experiment, right? This was no clear to me.

Author response

**Experimental durations were not identical due to practical limitations during experimentation. We argue that despite the dissimilar durations of both laboratory experiments, results are still comparable due to complete acclimation of intracellular osmolality (> 2 weeks) in response to different salinities (Neufeld et al., 1996). This has been addressed in the discussion (L502-514). Resultingly, we believe that in the context of the experiments presented here, short term exposures whether 1 or 2 months, would not result in significantly different effects of carbonate chemistry when both experiments are comparing the same species, sample population and life stage. Experimental durations were still long enough to detect a significant effect of seawater chemistry on calcification rates in juvenile bivalves.**

**The methods for measuring and calculated calcification rates during laboratory experiments has been clarified in the methods section (L275-287) of the revised manuscript. Explanations have been**

**given for the 2 time points when shell mass was quantified/calculated and an equation has been given (L285) for calculating calcification rates in each experiment.**

I understand that due to experimental limitations, the volume of the replicates had to be different. However, the final density (mussels ml-1) is too different to compare between both laboratory experiments. This is an issue to discuss in terms of denso-dependency potential effects on the results observed.

Author response

**The reviewer is right to point out that the number of mussels ml$^{-1}$ was not identical between both experiments. However, since mussels at the beginning of the Ca$^{2+}$ experiment had ~4 x the body mass of those at the beginning of the HCO$_3^-$ experiment (despite both cohorts being in the juvenile life stage), we argue a better metric for comparing both experiments is the total mussel biomass per L. This metric is also more applicable for identifying the metabolic effects on seawater chemistry between both experiments. This information has also been described in the results section (L354-356).**

**In the revised manuscript and supplementary document, Table S2 has been revised and clarified to reflect the key metrics to compare between both experiments. In the revised supplementary Table S2, the column 'No. animals per tank' now states the mean number of animals as an average of all treatments throughout each experiment, rather than presenting the initial number of animals in each experiment. This mean value has been used to calculate the mean body dry mass (BM) per litre, as opposed to using the initial number of animals to calculate this metric. Resultingly, mean BM throughout each experiment is now calculated as 13.2 mg L$^{-1}$ and 24.1 mg L$^{-1}$ for the Ca and HCO3 experiments, respectively.**

**The methods section of the revised manuscript has been rewritten to emphasise the steps taken to minimise the effects of differential mussel biomass densities between both experiments and the impacts on food availability between both experiments (L245-261).**

**We argue that monitoring of clearance rates during the experiment and ensuring saturated feeding conditions, combined with regular water chemistry monitoring and water changes ensuring minimum deviations in water chemistry resulting from biological activity, largely minimised the density-dependent effects across treatments and experiments.**

L190 and following. Suddenly, the authors show that a field experiment was also performed. However, nor the introduction or abstract is pointed out. In my opinion, this is a stronghold of this study. Please, try to incorporate this information in the last paragraph of the introduction, as well as in the abstract.

Author response

**The authors have re-ordered the storyline of the manuscript to first introduce the environment and field systems in the introduction, methods and results sections with descriptions and discussions of the laboratory experiments following after. The abstract now mentions the field study (L23-24 and L30-31) whilst the re-ordered introduction now introduces the field study on growth rates (L129-132) and the methods section describes the rational behind the field experiment (L141-142).**

About the field study, the authors collected the laboratory experimental mussels in Ahreenshoop, however, the authors also performed field experiments in the other two extra sites. I understand the objective of this, but this is not explained in the manuscript.

Author response

**The objective of comparing environmental conditions and calcification rates in the field have been described in the methods section (L141-142 and L161-163) of the revised manuscript. This was to follow the methodologies of previous studies and to cover the range of the steepest salinity gradient in the Southwest Baltic Sea. The reordering of the methods section also makes clearer the rational behind the choice of the 3 monitoring sites in this study.**

Authors, in the field experiment, estimated calcification rates from the reported SLCaCO3 relationship. I understand, that this is a unique relationship developed for a specific mussel population. However, after reading the introduction where authors pointed out that there are important differences along the salinity gradient. So, in my opinion, this relationship should be different among mussel populations. This could have important effects on the results. Indeed, why authors did not use the same methodology of the laboratory experiment, could improve the comparison of results.

Author response

**Unique SL-CaCO$_3$ relationships were developed for each of the 3 individual mussel populations and these relationships have already been presented in the supplementary material (Fig. S3 of revised supplementary material). This has been clarified in the methods section (169-174) with emphasis on the fact that separate population specific relationships exist. Direct measurement of CaCO$_3$ mass was not done for the mussels sampled in the field, as population specific SL-CaCO$_3$ relationships were already available and the number of mussels collected, and the frequency of collections would have necessitated a significant workload. The SL-CaCO$_3$ relationship for the Ahrenshoop population was used to calculate the initial CaCO$_3$ mass in lab experiments, and direct measurements of CaCO$_3$ mass were done at the termination of both experiments with identical methods. This has all been rewritten in the methods section (L275-289).**

Authors, in the laboratory experiment, show how they burned shells in order to eliminate organic matter from the shells in order to provide CaCO3 data and estimate calcification rates. Were there differences in the organic matter among populations? This is so important, as many previous studies have shown how marine calcifying organisms show different organic matter concentrations under different environmental conditions (lab or field). If authors could show this data would be very interesting to understand another potential factor affecting calcification rates. Indeed, shell organic matter (periostracum and inter, intra-crystalline organic matters) has a shell protection function under corrosive environments, but also as a substrate to favor crystallization and biomineralization processes.

Author response

**The reviewer makes an excellent and very important point here. In the laboratory experiments, initial CaCO$_3$ mass was calculated using a population specific SL-CaCO$_3$ relationship, whereas CaCO$_3$ mass in each treatment was measured using the muffle furnace method described in the methods section. Subsequently, there is no initial measure of shell organic (periostracum or shell matrix proteins) to calculate changes in shell organic content during the course of both experiments. The aim of this study was to investigate changes in inorganic CaCO$_3$ deposition with salinity and carbonate chemistry, rather than changes in total shell composition or shell organic content. Subsequently, this data is unfortunately not available from this study. Understanding how shell organic content may be modulated in Baltic mussel shells in light of predicted climate change is an important point for understanding the fate of calcifying Baltic mussels. A discussion on the importance of considering shell organic content and structure has been included (L541-550) with**

**reference to its implications for adaptive responses of shell formation in marine calcifiers as well as the energetic cost of shell production.**

How many times Chl-a was measured during the field experiment?

Author response

**The number of Chl-*a* measurements are as follows: Usedom: 25 data points; Ahrenshoop: 25 data**
**points; Kiel: 28 data points. All Chl-*a* monitoring occurred from January 2015 – December 2017. This information has be added to the header of Table 1 in the revised manuscript.**

I do not have major comments on the results section.

The discussion section is clear and also identified the major limitations of the study which is appreciated. Results are broadly discussed from many points of view, however as it was pointed out
above, I miss a discussion of other potential causes that could determine the results. Indeed, biomineralization processes not only incorporate CaCO3 precipitation but they incorporate the secretion of periostracum/shell organic matter which has an important function on biomineralization. Also, the entire biomineralization process is energetic expensive because of the secretion of these shell organic compounds. It would be nice the authors develop this idea as potential causes of the
results observed in order to complete the discussion section. If authors can show shell organic matter by treatment, this could help a lot to understand the results. Indeed, this could be a future research topic to develop. In addition, some methodological limitations of the study (pointed out above) such as denso-dependency are not discussed in the discussion.

Author response

**We have included a discussion on the potential for changes in shell organic content to impact the observed results in terms of increasing energetic costs of calcification (L541-550). Whilst all authors agree the importance of considering the impacts of density-dependent effects on the observed results, we have amended the manuscript to clearly show that measures were taken to minimise the impacts of differential biomass concentrations in experimental aquaria between treatments and**
**experiments. These include measures to ensure comparable food availability between both experiments (See Table S2 and L356-357), minimising the impacts of biological activity and food addition on pH and $pCO_2$ (Table 2 and 3; and Section 2.5).**

FIGURES AND TABLES.

I suggest changing the order of figures, first showing the environmental conditions of field study sites,
and then the results of calcification rates.

Author response

**We have re-ordered the presentation of the figures, tables and results section. The environmental monitoring and field study is presented and discussed first, followed by the results from the laboratory experiments. This reflects changes made to all sections of the revised manuscript to**
**reflect the change in the storyline order.**

In table 1, I noticed that there are important differences in pH conditions among experimental treatments, how could affect the calcification rates?

Author response

The authors agree that differences in pH are important to consider in the interpretation of the results. Whilst this was discussed in the original manuscript, the revised manuscript now contains a deeper discussion on the impacts of pH on calcification in the bicarbonate experiment (L427-443 and L521-527) and the potential impact of pH on mortality in this experiment (L525-527). This data on mortality has been presented in Fig S10, as suggested by reviewer 2. We would also like to emphasise that the impacts of pH are included in both SIR and ESIR calculations as [H⁺] (See equations 2 and 3) and as such, the impacts of pH on mussel calculated have not been disregarded (L534-536).

**References**

Neufeld, D. S., and Wright, S. H.: Response of cell volume in *Mytilus* gill to acute salinity change, J. Exp. Biol., 199, 473-484, 1996.

Riisgård, H. U., Pleissner, D., Lundgreen, K., LArse, P. S.: Growth of mussels *Mytlus edulis*, at algal (*Rhodomonas salina*) concentrations below and above saturation levels for reduced filtration rate, Mar. Biol. Res., 9, 1005-1017, DOI: http://dx.doi.org/10.1080/17451000.2012.742549, 2013.

**Reviewer 2**

Reviewer comment

This is a very nice manuscript from Sanders and collaborators dealing with the impact of seawater chemistry on mussel growth rates. The two experiments have been well designed in order to discriminate first the impact of salinity vs. carbonate chemistry changes and second salinity vs. calcium concentration changes. These experiments have been complemented by a field survey covering over 3 years. Monitoring of seawater physico-chemistry and mussel growth have been performed at 3 sites along a decreasing salinity gradient towards the Central Baltic. The study is well introduced although I agree with Reviewer#1 that paragraph L98-112 should be put up front. The methods are most of the time well explained and the results properly discussed. I have no doubts that this will be a nice contribution to the Biogeosciences journal. Congratulations to the authors!

That being said, I have a few concerns and questions that I would like the authors to answer:

1) I have to say that I was impressed on how many individuals you could fit in 2 L containers (1600 animals, small but still. . .). Since you did not consider a flow-through system and changed the water "only" 2 to 3 times weekly, I am really wondering how would change carbonate chemistry but also ammonium and oxygen concentrations between two water changes. Table 1 and 2 are not clear to me. Do these tables show the conditions in the experimental plastic aquaria and/or in the stock seawater? If measured in the aquaria, when were the samples taken? Before and/or after water changes? Were your aquaria aerated? I apologize in case I missed that in the text.

Author response

We have rewritten this aspect of the methods section to clearly state that water changes were conducted at frequencies designated by regular water chemistry monitoring during experiments. Water chemistry was measured before and after water changes in both experiments (L216 and L240) and results revealed pH, [HCO₃⁻] and [Ca²⁺] did not deviate by unacceptable levels (L351-354 and L357-359). The methods section now includes details on aeration in both experiments (L196 and L229). Tables 2 and 3 of the revised manuscript show mean water chemistry parameters in experimental aquaria during the experiment and this has also been clarified in the methods section and in the Table headings. Although oxygen saturation was not measured in experimental aquaria, we believe continuous aeration and regular water changes prevented severe reductions in oxygen levels during experiments. We have also addressed a similar concern raised by reviewer one in our response to reviewer 1's comment (L52-63 of this document).

**Ammonia excretion was not quantified, however given a conservatively high ammonia excretion rate in Baltic mussels of 20 µg $NH_4$ per gram dry weight $hr^{-1}$ (Tedengren & Kautsky 1987), the more biomass dense $HCO_3^-$ experiment (mean biomass 24 mg dry weight per litre) would have resulted in maximum ammonia concentration of 0.04 mg $L^{-1}$ immediately prior to a water change after 3 days accumulation. This value of 0.04 mg $L^{-1}$ is far below the $LC_{50}$ value for juvenile *Mytilus edulis* of 0.39**

**mg $NH_4$ $L^{-1}$ after 21 days exposure (Kennedy *et al.*, 2017). Subsequently we do not believe there to be any negative impacts of ammonia build-up in either experiments. These calculations have not been included in the revised manuscript, but it has been mentioned that water change frequency was sufficient to prevent respiratory build up of $CO_2$ and consequent drops in pH (L241-242 and L359-360).**

Reviewer comment

2) In Table S2, you report on a >50% mortality during the 70 days bicarbonate experiment, as well as an important range (10-75%) across treatments. Did you check whether you had some relationships between mortality rates and the imposed chemical changes? Did you replace the dead organisms? If not, what would be the effect on the amount of food available for each individual? Table S2 is not clear to me, what are these biomass data? At the start of the experiment? At the end? You mention on L173 that biomass per litre was comparable between the 2 experiments while I can read that it was 13.2 mg/L during the Ca2+ exp and 51.5 mg/L during the HCO3- exp, it does not seem comparable to me.

Author response

**Mortality rates did exhibit patterns in relation to seawater chemistry. Mortality rates were highest at low pH/[$HCO_3^-$] and slightly higher at low salinities (6) compared to higher salinities of 11 and 16. This data has been included in the supplementary material (Fig S10 of revised supplementary material) and the potential causes of this discussed in the Discussion section (L525-528). Dead organisms were not replaced but quantified and changes taken into account during experimentation**

**(L214-216).**

**The header of Table S2 has been adjusted to clearly state it presents the standard deviation of mortality rates across all treatments in the experiment. Feeding regimes were chosen in such a way to ensure saturated feeding conditions in all treatments (>10 000 phytoplankton cells $ml^{-1}$) and final calculations of food availability between experiments revealed cell number per unit biomass was**

**comparable between experiments (Table S2 and L354-355 of Results section). Table S2 has also been recalculated and now contains mean data over the entire experimental period, whereas it previously presented the number of animals at the beginning of the experiment. This updated value now shows mean dry body mass (BM) per $L^{-1}$ in both experiments to be 13.2 mg $L^{-1}$ and 24.1 mg $L^{-1}$ for the Ca and HCO3 experiments, respectively. These more accurate values show that biomass**

**densities were even more similar between experiments than presented in the originally submittd manuscript.**

**Table S2 presents data per replicate tank as a mean value over the entire experimental period. Both experiments are presented as a comparison, as well as the range of values within the $HCO_3^-$ experiment to highlight that biomass and biomass per ml varied by a higher degree *within* the $HCO_3^-$**

**experiment than *between* both experiments. The mean values for each experiment (13.2 mg/L and 24.1 mg/L) are therefore within the same order of magnitude and the revised manuscript now clearly states in L218-220, L241-242 and Table S2, that food availability and density-dependent impacts of biological activity on water chemistry were not dissimilar between both experiments**

Reviewer comment

3) I believe there is one aspect (maybe related to the point above) that should be discussed. During the first experiment (bicarbonate), mussels at salinity 6 did not grow much (maybe 5 microg/d; Fig. 2a). What is the reason why they grew much better during the second experiment (Fig. 2b) even when Ca2+ concentrations are below ambient levels (2.5 mmol/kg), reaching rates of 20 microg/d)? Is it due to the differences in terms of experimental design?

Author response

**The reviewer raises an interesting point here related to calcification rate differences between both experiments at a salinity of 6. The discussion has been adjusted to include reasoning behind these differences (L454-457 and L487-493). Possible explanations include maximum $[Ca^{2+}]$ in the calcium experiment being ca. 0.2 mmol $kg^{-1}$ below threshold values and subsequently masking the impacts**
**of low salinity on calcification in that experiment. Potential genetic differences between cohorts and minor differences in mean animal sizes between experiments have also been discussed in the revised manuscript (L494-496).**

Reviewer comment

4) As such, I do not believe that trying to fit any model to all data points (pooled from the two
experiments) makes much sense (Fig. S4 and S5, but also Fig. 3). At least for a better view on the data, you should identify the dots depending on the experiment and salinity levels.

Author response

**The authors agree and have altered Figures 6, S4 and S5 to label data points based on experiment and salinity levels for increased clarity.**

Reviewer comment

5) It seems that you over-determined carbonate chemistry during the field survey by measuring pH, CT and AT. It is not clear to me if AT data showed (i.e. Fig. 7) are the ones measured or derived from pH and CT, maybe to clarify. Finally, how do computed AT and measured AT compare? This could be a nice way to identify DOC contribution no?

Author response

**$A_T$, $C_T$ and pH were all determined from field samples, however only pH and $C_T$ were used to calculate other carbonate chemistry parameters due to the potential impacts of dissolved organic matter on measured $A_T$, as the reviewer mentioned. Measured $A_T$ data is not shown in Fig. 3 of the revised manuscript, but rather the subsequent carbonate chemistry parameters calculated from field**
**measurements of pH and $C_T$. The reviewer makes an interesting point in comparing measured $A_T$ and calculated $A_T$ from $C_T$ and pH. However, the potential contribution of DOM towards $A_T$ is complex, as this organic alkalinity contribution ($A_{org}$) is not a linear function of DOC, but rather dependent on various parameters such as pH. Previous studies found that $A_{org}$ ranged from 22–58 µmol $kg^{-1}$, and developed a first mechanistic understanding of how this contribution relates to the**
**amount and nature of DOM, as well as seawater carbonate chemistry (Kuliński et al., 2014). Simply**

comparing our measured $A_T$ and calculated $A_T$ would not help us to better understand the $A_{org}$ contribution to alkalinity, while a detailed analysis of this contribution is well beyond the scope of this paper. Thus, we prefer not to include the suggested comparison.

**References**

Kennedy, A. J., Lindsay, J. H., Biedenbach, J. M., Harmon, A. R.: Life stage sensitivity of the marine mussel *Mytilus edulis* to ammonia, Environ. Toxicol. Chem. 36, 89-95, DOI: 10.1002/etc.3499, 2017.

Kuliński, K., Schneider, B., Hammer, K., Machulik, U. and Schulz-Bull, D.: The influence of dissolved organic matter on the acid–base system of the Baltic Sea, J. Mar. Syst., 132, 106–115, https://doi.org/10.1016/j.jmarsys.2014.01.011, 2014.

Tedengren, M., Kautsky, N.: Comparative study of the physiology and its probably effect on size in blue mussels (*Mytilu edulis* L.) from the North Sea and the northern Baltic Proper, Ophelia, 25, 147-155, DOI: 10.1080/00785326.1986.10429746, 1986.